# Investigating the trade-off between folding and function in a multidomain Y-family DNA polymerase

Xiakun Chu[1], Zucai Suo[2], Jin Wang[1]*

[1]Department of Chemistry, State University of New York at Stony Brook, New York, United States; [2]Department of Biomedical Sciences, College of Medicine, Florida State University, Tallahassee, United States

**Abstract** The way in which multidomain proteins fold has been a puzzling question for decades. Until now, the mechanisms and functions of domain interactions involved in multidomain protein folding have been obscure. Here, we develop structure-based models to investigate the folding and DNA-binding processes of the multidomain Y-family DNA polymerase IV (DPO4). We uncover shifts in the folding mechanism among ordered domain-wise folding, backtracking folding, and cooperative folding, modulated by interdomain interactions. These lead to 'U-shaped' DPO4 folding kinetics. We characterize the effects of interdomain flexibility on the promotion of DPO4–DNA (un)binding, which probably contributes to the ability of DPO4 to bypass DNA lesions, which is a known biological role of Y-family polymerases. We suggest that the native topology of DPO4 leads to a trade-off between fast, stable folding and tight functional DNA binding. Our approach provides an effective way to quantitatively correlate the roles of protein interactions in conformational dynamics at the multidomain level.

## Introduction

Our understanding of protein folding has been deepened by intensive experimental, theoretical, and computational studies focused on single-domain proteins or isolated domains of multidomain proteins (*Jackson, 1998*). However, it is widely recognized that throughout all three kingdoms of life, proteins occur predominately in multidomain forms (*Apic et al., 2001*; *Ekman et al., 2005*). As their name indicates, multidomain proteins consist of more than one structural building unit, or domain (*Teichmann et al., 1999*). Domains themselves have a strong tendency to fold (*Murzin et al., 1995*), but although there is high structural modularity in a multidomain protein (*Han et al., 2007*), the folding of a multidomain protein usually takes a more complex form than a simple sum of folding of individual domains (*Levy, 2017*). The key component in the folding of a multidomain protein is the interaction of domain interfaces or linkers, which have been found to play nonuniversal roles in modulating folding stability (*Fast et al., 2009*; *Bhaskara et al., 2013*; *Vishwanath et al., 2018*), cooperativity (*Batey et al., 2005*) and kinetics (*Osváth et al., 2005*; *Batey et al., 2006*; *Batey and Clarke, 2006*).

Efficient folding of a multidomain protein is vital not only for providing structural scaffolds for biological function (*Vogel et al., 2004*), but also for preventing misfolding (*Strucksberg et al., 2007*). Multidomain proteins, which often possess significant domain interfaces, are more prone to aggregation during folding processes than single-domain proteins (*Han et al., 2007*; *Borgia et al., 2011*). It has been suggested that in vivo, multidomain proteins can undergo co-translational folding (*Fedorov and Baldwin, 1997*), where each domain folds sequentially one-by-one during protein synthesis from the ribosome (*Netzer and Hartl, 1997*; *Frydman et al., 1999*). Likewise, a 'divide-and-conquer' scenario has been proposed for in vitro multidomain protein folding, where all domains

*For correspondence:
jin.wang.1@stonybrook.edu

**Competing interests:** The authors declare that no competing interests exist.

fold independently, followed by coalescence of neighbors (*Wang et al., 2012a*). In both of these folding scenarios, independent domain folding plays an essential role and is deemed to drive the global folding. At the same time, the role of domain coupling in multidomain protein folding appears to be important, but its complexity means that a definitive conclusion cannot easily be drawn (*Batey et al., 2008*). A recent computational study of a two-domain serpin elucidated the critical role of the functional binding-related reactive center loop (RCL) in the folding of the protein to distinct structures (*Giri Rao and Gosavi, 2018*). Folding of the serpin to the metastable active structure, where the RCL is present as an intradomain segment, is faster than folding to the stable latent structure, where the RCL is involved in extensive interactions between domains. Other work using a similar model, however, indicated that removal of interdomain interactions had little effect on the folding cooperativity of a three-domain adenylate kinase (*Giri Rao and Gosavi, 2014*). Using statistical mechanical models, Sasai and co-workers investigated a variety of multidomain proteins and their circular permutants. Their results showed that domain connectivity and interactions in multidomain proteins determine folding pathways, cooperativity, and kinetics (*Itoh and Sasai, 2008*; *Inanami et al., 2014*). However, at present, a unified perspective on the role of interdomain interactions is still missing. Addressing this issue is an important avenue in studies of multidomain protein folding.

From a structural perspective, the effects of neighboring domains in terms of interdomain interactions are fundamental for generating and stabilizing the correct multidomain folds (*Jones et al., 2000*; *Bhaskara and Srinivasan, 2011*). On the other hand, overwhelming interdomain interactions may distort domains from the structurally folded units, reducing the efficiency that comes with domain-wise folding. To achieve a 'speed–stability' balance, a multidomain protein may optimize the strength of interdomain interactions to simultaneously guarantee efficient folding through the 'divide-and-conquer' folding mechanism and successful formation of functional structures with the aid of stabilization from the domain interface. Usually, the relatively weak interdomain interactions trigger domain motions in multidomain proteins, making a pivotal contribution to protein function (*Bennett and Huber, 1984*; *Schulz, 1991*; *Miyashita et al., 2003*). As is now widely recognized, the native folds of proteins may exhibit a certain degree of frustration in favor of functional state switching (*Ferreiro et al., 2007*; *Ferreiro et al., 2014*; *Whitford and Onuchic, 2015*). Therefore, the energetics of interdomain interactions in multidomain proteins may be evolutionarily optimized for making the trade-off between fast, stable folding and efficient, tight substrate binding (*Bigman and Levy, 2020*). However, it is still unclear how a multidomain protein manages the intricate balance among its interactions to allow simultaneous folding and function. Here, we aim to answer this fundamental question through a computational study of the folding and DNA-binding processes of *Sulfolobus solfataricus* DNA polymerase IV (DPO4), a prototype Y-family DNA polymerase.

Akin to the other Y-family polymerases, DPO4 consists of a polymerase core with a right-handed architecture, including finger (F), palm (P), and thumb (T) domains, as well as a little figure (LF) domain that is connected to the polymerase core by a flexible linker (*Figure 1A*; *Ling et al., 2001*). Structural analysis has shown that many more intradomain contacts than interdomain contacts are present in the apo form of DPO4 (*Wong et al., 2008*; *Appendix 2—table 1*). Thermal unfolding experiments have indicated the existence of one intermediate state (*Sherrer et al., 2012*), which is probably formed by the unfolding of the linker interactions with the domains. These features imply that the four domains of DPO4, though differing in size and topology, are prone to fold independently. Binding of DPO4 to DNA is an essential step in nucleotide incorporation (*Fiala and Suo, 2004*; *Wong et al., 2008*). Structural comparisons of DPO4 in apo form and DNA binary form have revealed that significant rotation and translation of the LF domain occur during DNA binding, while the domain structures remain unchanged (*Wong et al., 2008*). This 'open-to-closed' conformational transition in DPO4 is pivotal to the formation of a high-affinity DPO4–DNA complex prior to nucleotide binding and incorporation (*Fiala and Suo, 2004*; *Sherrer et al., 2009*). Previous experimental and simulation studies have suggested that the linker plays an important role in this conformational transition (*Xing et al., 2009*; *Sherrer et al., 2012*; *Chu et al., 2014*). Nevertheless, a thorough investigation of the roles played by DPO4 domain interactions in the modulation of DNA binding and protein conformational dynamics is still lacking. More importantly, it remains unclear how the folding and DNA binding of DPO4 can be optimized by the interactions in DPO4.

Here, we use structure-based models (SBMs) with a comprehensive procedure for parameterizing the strengths of intra- and interdomain interactions to investigate the folding and DNA-binding

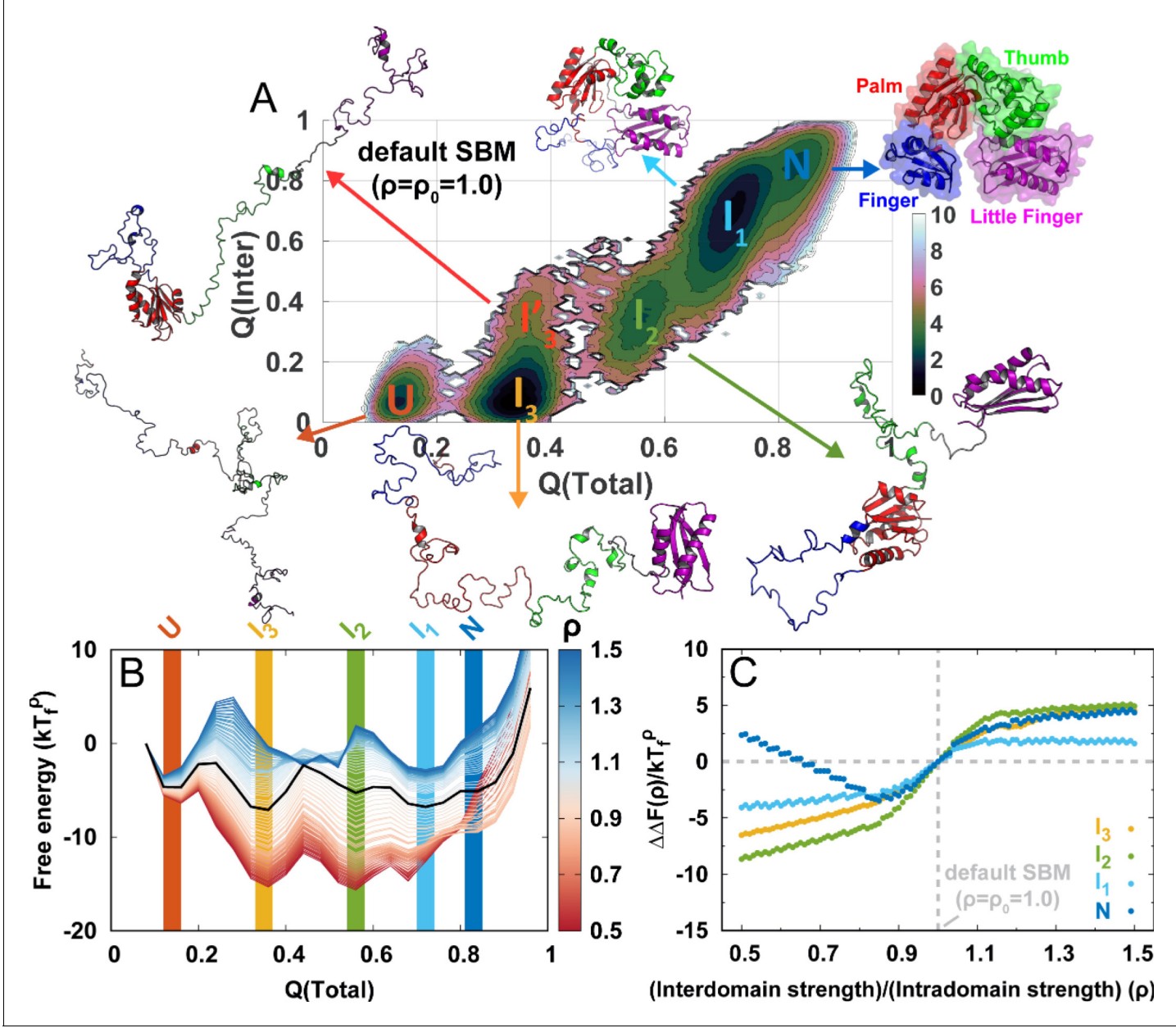

**Figure 1.** DPO4 folding thermodynamics. (A) The 2D folding free energy landscape of DPO4 projected onto the fractions of total ($Q(\mathrm{Total})$) and interdomain ($Q(\mathrm{Inter})$) native contacts for the default SBM parameter $\rho_0$ at the folding temperature $T_f^{\rho_0}$. There are six folding states identified on the free energy landscape, which are denoted by $U$, $I_3$, $I_3'$, $I_2$, $I_1$, and $N$, with corresponding typical DPO4 structures shown. The all-atom representations of DPO4 were reconstructed based on the $C_\alpha$ structures from the simulations (*Rotkiewicz and Skolnick, 2008*). Domains of DPO4 are labeled with different colors: the finger domain (F, blue, residues 11–77), the palm domain (P, red, residues 1–10 and 78–166), the thumb domain (T, green, residues 167–229), the little finger domain (LF, purple, residues 245–341), and the flexible linker (gray, residues 230–244), which connects the T and LF domains. (B) The 1D folding free energy landscape of DPO4 projected onto $Q(\mathrm{Total})$ for different values of the ratio of interdomain to intradomain native contact strength, denoted by $\rho$ and ranging from 0.5 to 1.5 (indicated by different colors). It is worth noting that the $I_3$ and $I_3'$ states cannot be distinguished by $Q(\mathrm{Total})$. The black line corresponds to the free energy landscape for the default SBM parameter $\rho_0$. We set the zeros of the free energies at the lowest $Q(\mathrm{Total})$ detected from the simulations. (C) Change in stability of each folding state for $\rho$ relative to that for $\rho_0$ at the corresponding folding temperature $T_f^\rho$ according to the expression $\Delta\Delta F(\rho)^S = \Delta F(\rho)^S - \Delta F(\rho_0)^S$, where $S$ represents the folding state ($I_3$, $I_2$, $I_1$, or $U$) and $\Delta F(\rho)^S$ is the stability of the folding state $S$ at $\rho$. $\Delta F(\rho)^S$ is calculated as the free energy difference between the folding state $S$ and the unfolded state $U: \Delta F(\rho)^S = F(\rho)^S - F(\rho)^U$, where $F$ is the free energy obtained from (B).

The online version of this article includes the following figure supplement(s) for figure 1:

**Figure supplement 1.** Native contact maps of DPO4 for the folding states $U$, $I_3$, $I_3'$, $I_2$, $I_1$, and $N$.

processes of DPO4. We find a monotonic increase in folding stability led by a decrease in the interdomain interaction strengths for all intermediate states during folding but not the folded states. This further underlines the importance of interdomain connections in maintaining the correct fold. Interestingly, we find that strengthening the interdomain interactions can result in an increase in folding cooperativity and the chances of backtracking (*Gosavi et al., 2006*). The interplay among folding stability, backtracking, and cooperativity leads to a 'U-shaped' interdomain interaction-dependent kinetics. Finally, we quantitatively characterize the role of a flexible domain interface in accelerating the fast DNA (un)binding to DPO4, which probably promotes DNA lesion bypass during DNA synthesis undertaken by a Y-family enzyme. Our results suggest that the topology and interactions of DPO4 have been optimized to achieve its fast folding and tight DNA binding, plausibly by evolutionary pressure. Our systematic investigation of the interactions in a multidomain protein provides a mechanistic understanding of the relationship between protein folding and function at the multidomain level and offers useful guidance for multidomain protein engineering.

## Results

### Effects of interplay between intra- and interdomain interactions on DPO4 folding thermodynamics

SBMs are simplified models based on energy landscape theory (*Bryngelson et al., 1995*). They include only the interactions in the protein native structure and have proven successful in capturing protein folding mechanisms (*Clementi et al., 2000*). The essential assumption made by SBMs is that native contacts should determine the protein folding mechanism, which has been confirmed by all-atom simulations (*Best et al., 2013*). Further systematic comparisons between coarse-grained SBMs and all-atom simulations have also shown high consistency in extensive predictions for the folding of proteins with diverse native topologies (*Hu et al., 2017*). These features, together with the reliability and computational affordability of SBMs, indicate that they are attractive models for investigating protein folding, especially in the case of complex large proteins. Therefore, in this study, we use a coarse-grained SBM to simulate the folding of DPO4.

To improve the sampling, we use replica-exchange molecular dynamics (REMD) to explore the thermodynamics of DPO4 folding (*Sugita and Okamoto, 1999*). We apply the default parameter in the SBM, which sets the same strength for all the native contacts ($\rho = \rho_0 = 1.0$; details are presented in Materials and methods). We project the folding free energy landscape onto the fraction of native contacts, $Q$, which has been recognized as a good reaction coordinate for describing folding of the single-domain proteins with two-state manner by means of SBMs (*Best and Hummer, 2005*; *Cho et al., 2006*). In addition, there are previous studies using $Q$ to describe the folding of various multidomain proteins (*Li et al., 2012*; *Giri Rao and Gosavi, 2014*; *Inanami et al., 2014*; *Tanaka et al., 2015*; *Giri Rao and Gosavi, 2018*), supporting the validity of $Q$ as a reaction coordinate for describing DPO4 folding. However, more precise and detailed description of the multidomain protein folding process may require the involvement of more reaction coordinates. Here, we use the interdomain and total contact $Q$ ($Q(\text{Inter})$ and $Q(\text{Total})$) to describe DPO4 folding (*Figure 1A*). The 2D free energy landscape reveals a complex DPO4 folding process with multiple intermediate states. The existence of these intermediate states in DPO4 folding is in good agreement with the results of temperature-induced unfolding experiments, which revealed more than one transition during the unfolding of DPO4 (*Sherrer et al., 2012*).

To see how DPO4 folds, we analyze the structures of DPO4 in the (meta)stable states indicated by free energy landscape minima from contact maps (*Figure 1—figure supplement 1*). This enables us to gain insight into domain and interface formation in DPO4 during folding. We see that $U$ and $N$ are the completely unfolded and fully folded states, with little and fully formed native contacts in DPO4, respectively; $I_1$ is an intermediate folding state in which DPO4 has only an unfolded F domain; DPO4 in the $I_2$ state further unfolds the T domain from $I_1$; in its $I_3$ and $I_3'$ states, DPO4 possesses a similar $Q(\text{Total})$ but has a different $Q(\text{Inter})$. From the contact maps, we find that there is only one folded domain in DPO4 in the $I_3$ or $I_3'$ state: the LF domain in $I_3$ and the P domain in $I_3'$. Folding of the P domain can partly stabilize the formation of P–T and P–F interfaces within the DPO4 polymerase core (*Silvian et al., 2001*; *Trincao et al., 2001*), leading to an increase in $Q(\text{Inter})$. By contrast, folding of the LF domain does not trigger any interdomain formation, probably

because the LF domain is separated by the flexible linker, far from the DPO4 core (**Boudsocq et al., 2001**; **Ling et al., 2001**). Overall, DPO4 folding complies with a 'divide-and-conquer' framework (**Wang et al., 2012a**). Such a folding mechanism obtained from the SBM with default parameter $\rho_0$ (the same native contact strength) is probably a consequence of the DPO4 topology (**Baker, 2000**), which exhibits many more intradomain contacts than interdomain ones (**Appendix 2—table 1**).

To see how a change in the balance between intra- and interdomain interactions in DPO4 may influence folding, we modulate the relative strength of these interactions in the SBM through the ratio $\rho = \epsilon_{\text{Inter}}/\epsilon_{\text{Intra}}$, where $\epsilon_{\text{Intra}}$ and $\epsilon_{\text{Inter}}$ are the strengths of the native contacts for intra- and inter-domain interactions, respectively. In practice, this is implemented by changing only $\epsilon_{\text{Inter}}$ while keeping $\epsilon_{\text{Intra}}$ and the other parameters to the default as they are in a homogeneously weighted SBM (**Clementi et al., 2000**). Using a reweighting method (**Cao et al., 2016**; **Li et al., 2018**), we quantify the free energy landscapes for DPO4 folding by different values ρ. In **Figure 1B**, we see that changing ρ within a moderate range (0.7–1.3) does not alter the multistate characteristics of DPO4 folding, since the values of $Q(\text{Total})$ for all the (meta)stable states remain almost the same as those at $\rho_0$. Further decreasing or increasing ρ can distort the free energy landscape from that at $\rho_0$. In general, this involves an alteration of the DPO4 folding mechanism when the strength of interdomain interactions deviates significantly from the default value.

Change in interactions in proteins, for example, by means of the mutations, pH, and denaturants, can affect folding stability. Likewise, modulation of the strength of the intra- and interdomain interactions may change the stability of different states during DPO4 folding. In **Figure 1C**, we can see that all the intermediate states of DPO4 exhibit monotonic decreases in folding stability as the interdomain interaction strengths increase ($\Delta\Delta F$ increases as ρ increases), although with different magnitudes. Since DPO4 in its intermediate states forms more intradomain contacts than interdomain ones, increasing the weight of interdomain interactions in the SBM is expected to weaken the folding stability in the intermediate states. DPO4 in the $I_2$ state possesses two large-sized P and LF domains, which are distal structural units and do not have interactions in between (**Appendix 2—table 1**). Strengthening the interdomain interactions relatively weakens the intradomain interactions, thus leading to destabilization of the $I_2$ state, which exhibits the most significant decrease among the intermediates as ρ increases. DPO4 in the $I_1$ state is stabilized by both intra- and interdomain interactions (**Figure 1—figure supplement 1**). Thus, increasing the interdomain interaction strength has less of a destabilization effect than that on the $I_3$ state, where only one domain in DPO4 is folded. An interesting nonmonotonic ρ-dependent behavior is observed in the $N$ state, where weakening the interdomain interactions (decreasing ρ below ~0.85) can instead lead to a decrease in folding stability. This is probably because the structures of DPO4 in the $N$ state are maintained by cooperation between intra- and interdomain interactions. Increasing the relative weight of intradomain interactions in the SBM (decreasing ρ) cannot always increase the stability of the $N$ state, since the fragile interdomain interactions may not be able to form domain interfaces within the DPO4 native structure.

## Effects of interplay between intra- and interdomain interactions on the DPO4 folding mechanism

The 2D free energy landscape for DPO4 folding with the default SBM parameter $\rho_0$ indicates that there are at least two potential folding pathways that go separately from $I_3$ and $I'_3$ to $I_2$ and then to $I_1$ (**Figure 1A**). Increasing and decreasing the strength of the interdomain interactions appear to enhance one of these two pathways (**Figure 2—figure supplement 1**). Very weak ($\rho = 0.5$) or very strong ($\rho = 1.5$) interdomain interactions can shift DPO4 folding completely to one pathway (**Figure 2A** and **Figure 2B**), resonating with the findings from the 1D free energy landscape that a very high or low ρ can change the DPO4 folding mechanism. We further calculate the averaged $Q(\text{Inter})$ and $Q(\text{Total})$ and interestingly find that there are two regions exhibiting an increase followed by a decrease in $Q(\text{Inter})$ as $Q(\text{Total})$ increases (**Figure 2C**). This observation may be a sign of folding 'backtracking,' which involves the formation, breaking, and refolding of a subset of native contacts as the protein proceeds from the unfolded to the folded state (**Gosavi et al., 2006**). As the REMD simulations have broken the kinetic connectivity of the folding trajectory, we perform additional kinetic simulations at constant temperature to assess the backtracking rigorously. Although a temperature ($0.96T_f$) lower than the folding temperature is used in the kinetic simulations with the aim of generating a sufficient number of folding events, we still observe the backtracking within

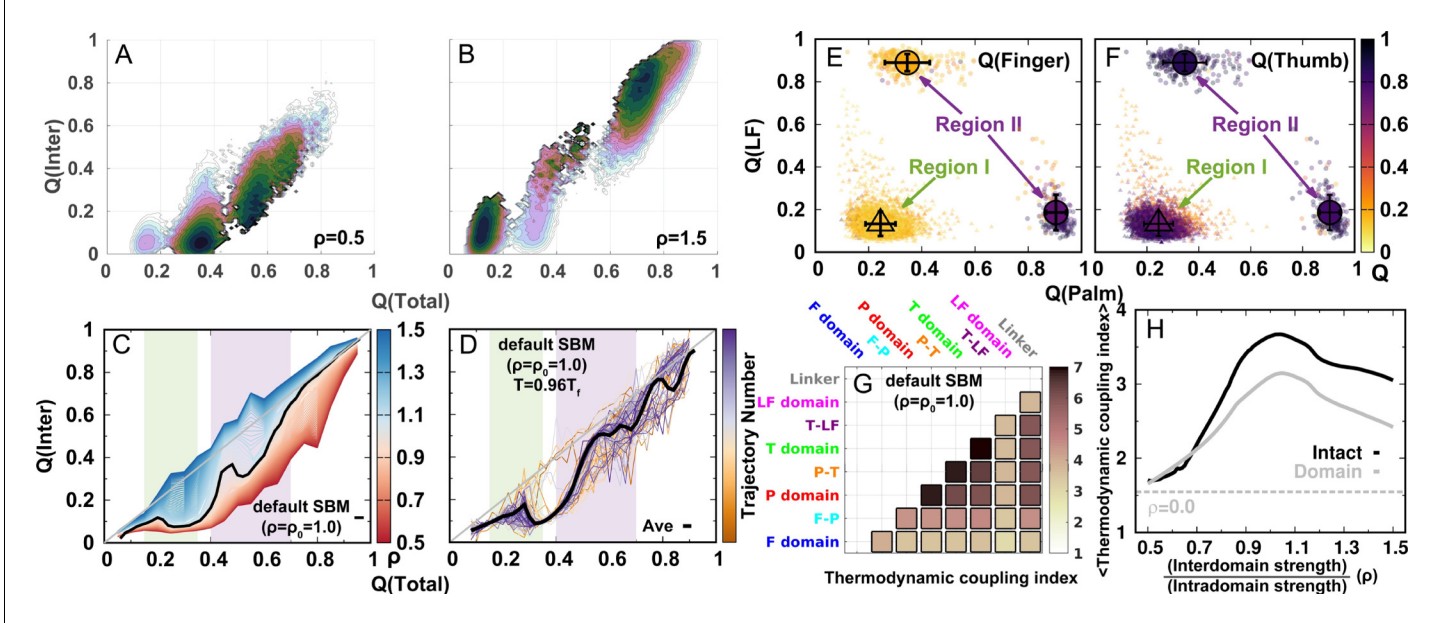

**Figure 2.** DPO4 folding free energy landscapes, backtracking, and folding cooperativity for different values of ρ. (a, B) 2D folding free energy landscapes of DPO4 for (A) $\rho = 0.5$ and (B) $\rho = 1.5$. (C) Averaged $Q(\mathrm{Inter})$ versus $Q(\mathrm{Total})$ for different values of ρ at the corresponding folding temperature $T_f^\rho$ calculated from the 2D free energy landscapes. The black line corresponds to the result with the default parameter $\rho_0$. The two shaded regions show the increase followed by the decrease in $Q(\mathrm{Inter})$ with increasing $Q(\mathrm{Total})$ for ρ close to its default value $\rho_0$, indicating possible backtracking during DPO4 folding. (D) Averaged $Q(\mathrm{Inter})$ versus $Q(\mathrm{Total})$ at a temperature $0.96T_f$ and for the default parameter $\rho_0$, as calculated from the kinetic simulations. The colored lines are the observations for the individual folding events in all the simulations (a total number of 100), and the black line indicates the average. The domain structure in DPO4 in the backtracking regions is described by the fractions of native contacts $Q(\mathrm{LF})$, $Q(\mathrm{Palm})$, $Q(\mathrm{Finger})$, and $Q(\mathrm{Thumb})$, the last two of which are shown in (E) and (F), respectively. There are two separate regions in backtracking region II, implying two distinct DPO4 structures. (G) Thermodynamic coupling index (TCI) for the default parameter $\rho_0$. (H) Dependence of the mean thermodynamic coupling index (MTCI) on ρ. The black and gray lines show the MTCI of the intact DPO4 and the four domains in DPO4, respectively. The dashed line indicates the MTCI calculated from the independent folding simulations of the four domains of DPO4, reminiscent of the extreme case with $\rho = 0$.

The online version of this article includes the following figure supplement(s) for figure 2:

**Figure supplement 1.** 2D free energy landscapes for DPO4 folding with different values of ρ.

**Figure supplement 2.** Averaged $Q(\mathrm{Inter})$ and $Q(\mathrm{Total})$ at a temperature $0.96T_f$ for different values of ρ.

**Figure supplement 3.** Representative DPO4 folding trajectory to illustrate backtracking.

**Figure supplement 4.** Number of backtracking observed for different values of ρ.

**Figure supplement 5.** Native contact maps and structures of DPO4 at backtracking.

**Figure supplement 6.** Melting curve of each domain/interface and two-state sigmoidal fitting.

**Figure supplement 7.** TCI for different values of ρ.

many of the individual folding trajectories (*Figure 2D*, *Figure 2—figure supplement 2*, and *Figure 2—figure supplement 3*). Further analysis of the number of backtracking shows that backtracking is most probable when the interdomain interaction strength is higher than the default value in the SBM (*Figure 2—figure supplement 4*).

To address the origins of backtracking, we extract all the DPO4 structures where $Q(\mathrm{Inter})$ shows a local maximum versus $Q(\mathrm{Total})$ in the two backtracking regions and perform a structural analysis by calculating $Q$ for each domain in DPO4. We find that DPO4 in backtracking region I ($Q(\mathrm{Total}) = 0.15$–$0.35$) probably has a folded T domain, while the other domains remain unfolded (bottom left in *Figure 2E* and *Figure 2F*). Since region I is located between the $U$ and $I_3$ states, we deduce that backtracking is caused mainly by folding and subsequent unfolding of the T domain. On the other hand, DPO4 in backtracking region II ($Q(\mathrm{Total}) = 0.4$–$0.7$) has mainly either folded T and LF domains (top left in *Figure 2E* and *Figure 2F*) or a folded core formed from the F, T, and P domains (bottom right in *Figure 2E* and *Figure 2F*) during the transition from the $I_3$ or $I_3'$ state to the $I_2$ and $I_1$ states. The bimodal distribution of DPO4 structures in region II indicates that

backtracking exists within both of the two folding pathways. Therefore, we suggest that the back-tracking in DPO4 folding is led by unstable and fast domain folding and unfolding of the small-sized F and T domains (*Figure 2—figure supplement 5*).

Folding cooperativity measures how synchronous residues in a protein form native-like configurations during folding. It dictates the folding mechanisms of the single-domain proteins in two-state 'all-or-none' folding, folding with intermediates, and downhill folding (*Muñoz et al., 2016*). Here, we apply a thermodynamic coupling index (TCI) to quantify the folding cooperativity of the domains and interfaces in DPO4. TCI is a measure of the similarity between a pair of intra- and interdomain melting curves during DPO4 unfolding (*Sadqi et al., 2006*; *Sborgi et al., 2015*). A large (small) TCI leads to a high (low) degree of synchronous folding between the domains/interfaces and thus a high (low) folding cooperativity (TCI is defined in Materials and methods). In *Figure 2G*, we can see that there is high cooperative folding in the DPO4 core. This is consistent with the experimental evidence that the conserved DPO4 core is a stable structural unit that unfolds cooperatively (*Sherrer et al., 2012*). When the interdomain interaction is weakened, DPO4 folding cooperativity decreases (*Figure 2H*). This trend is confirmed both when all domains/interfaces are taken into account and when only domains are considered. We also perform independent REMD folding simulations for individual domains of DPO4. This corresponds to the extreme case in which the interdomain interactions of the four domains of DPO4 are completely removed, reminiscent of the case $\rho = 0$. We observe an increase in the mean TCI (MTCI) as $\rho$ increases from 0 to 0.5, indicating that the presence of interdomain interactions enhances cooperative folding among the four domains of DPO4. Interestingly, the relation between $\rho$ and MTCI is not entirely monotonic, since MTCI decreases slightly after $\rho$ increases beyond ~1.0. This may be due to the following two reasons. (1) Increasing $\rho$ tends to separate the folding curves of the unstable F domain and its associated F–P domain from the other parts of DPO4 (*Appendix 3—figure 7*). Removing the unfolding curves of the F domain and F–P interface when calculating MTCI can lead to a significant increase in its value as well as in the value of $\rho$ at which MTCI reaches its maximum (*Appendix 3—figure 8*). (2) Increasing $\rho$ decreases the relative intradomain contributions in an SBM and may have different magnitudes of destabilization of the folding of different domains. Such a decrease in MTCI with increasing $\rho$ can be found in applications to independent folding of individual domains with a reweighting method (*Appendix 3—figure 9*). Overall, we find that a moderate increase in interdomain interaction strength can significantly increase the folding cooperativity of DPO4 (for values of $\rho$ in the range 0.5–1.0).

## Effects of interplay between intra- and interdomain interactions on DPO4 folding kinetics

Previous studies have suggested that the topology of a protein's native structure is an important determinant of its folding rate (*Plaxco et al., 1998*; *Koga and Takada, 2001*). Interaction heterogeneity, which originates from the amino acid sequence, has also been recognized as affecting folding kinetics (*Islam et al., 2004*; *Calloni et al., 2003*; *Szczepaniak et al., 2019*). To see how the energetic factor in terms of intra- and interdomain interactions in DPO4 affects the folding rate, we perform 100 independent kinetic simulations starting from different unfolded configurations at room temperature $T_r$ for each $\rho$. In practice, we use the time at which Q reaches 0.75, termed as the first passage time (FPT) to represent the folding rate. We observe a 'U-shaped' $\rho$-dependent mean FPT (MFPT) behavior for DPO4 global folding kinetics (*Figure 3A*). The value of $\rho$ at which DPO4 global folding is fastest is 0.8, which is lower than the default parameter $\rho_0$. There are distinct effects of $\rho$ on the kinetics of domain and interface folding. Intuitively, weakening interdomain interactions (decreasing $\rho$) should promote domain folding (*Figure 3B*), since the increased folding stability of individual domains resulting from a relatively strengthening of intradomain interactions should accelerate the folding process (*De Sancho et al., 2009*; *Garbuzynskiy et al., 2013*). On the other hand, interdomain folding exhibits 'U-shaped' behavior with $\rho$, similar to global DPO4 folding. When interdomain interactions are weak ($\rho$ in the range 0.5–0.9), the formation of interfaces between domains in DPO4 consumes more time than domain folding and thus is the rate-limiting step. Increasing $\rho$ beyond 0.9 slows down interdomain formation. During DPO4 folding, domain interfaces use the folded domains as templates to proceed further. When $\rho$ is high ($\geq$1.0), corresponding to the case of weak intradomain interactions, domain folding is slow and may become the rate-limiting step (*Figure 3B*). Therefore, we investigate switching of the bottleneck for DPO4 folding between

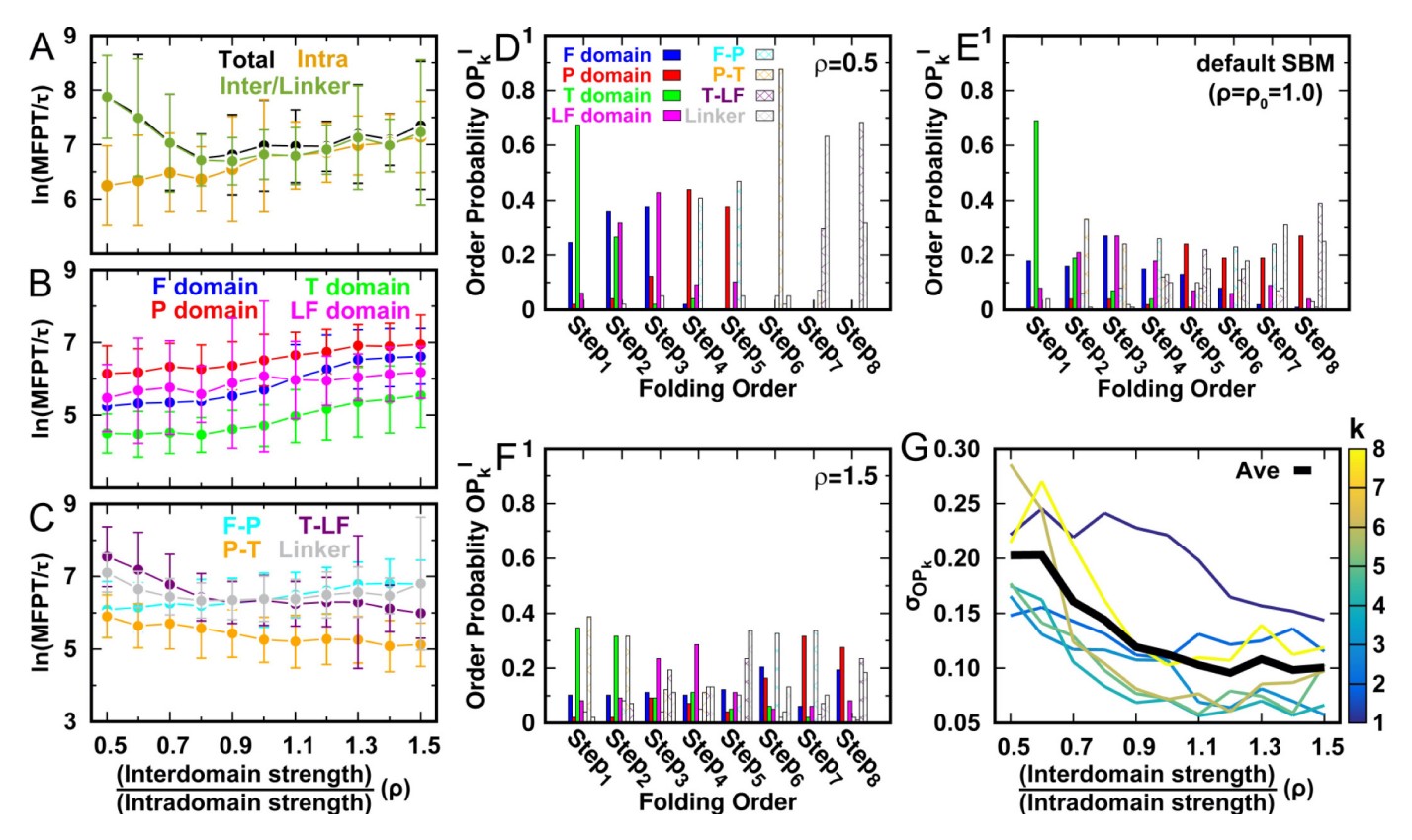

**Figure 3.** DPO4 folding kinetics at room temperature $T_r$. (A–C) Kinetic rates quantified by the mean first passage time (MFPT) for (A) intradomain, interdomain, and total folding, (B) individual intradomain folding, and (C) interdomain formation. MFPT is in units of $\tau$, which is the reduced time unit used in our simulations. Error bars represent the standard deviations at the corresponding MFPT. A folding is defined as when the corresponding $Q$ is higher than 0.75. (D–F) Folding order probability $\mathrm{OP}_k^I$ of individual intra- and interdomain occurs during a successful DPO4 folding event with (D) $\rho = 0.5$, (E) the default parameter $\rho = \rho_0 = 1.0$, and (F) $\rho = 1.5$. $k$ is the order index of the folding step and $I$ is the index of the domain/interface of DPO4. There are eight domains/interfaces in DPO4, and thus eight folding steps are defined. (G) Standard deviation of $\mathrm{OP}_k$ from its mean $\langle \mathrm{OP}_k \rangle = \frac{1}{8} \sum_I \mathrm{OP}_k^I \equiv \frac{1}{8}$ at folding step $k$ for different values of $\rho$. The black line shows the average $\sigma_{\mathrm{OP}_k}$ versus $\rho$.

The online version of this article includes the following figure supplement(s) for figure 3:

**Figure supplement 1.** $\mathrm{OP}_k^I$ for different values of $\rho$.

**Figure supplement 2.** DPO4 folding kinetics at a temperature $0.96 T_f$.

**Figure supplement 3.** $\mathrm{OP}_k^I$ for different values of $\rho$ at a temperature $0.96 T_f$.

interface formation and domain folding through modulation of the intra- and interdomain interaction strengths.

In contrast to domain folding, we find that formation of domain interfaces and the linker show different $\rho$-dependent behaviors (**Figure 3C**). The formation of the T–LF interface and the linker are significantly accelerated when $\rho$ increases from 0.5 to 0.8. These two interdomain structures are critical to DPO4–DNA functional binding, where a large-scale 'open-to-closed' conformational transition with rearrangement of the T–LF interface and the linker has been observed between apo-DPO4 and DNA-binding DPO4 (**Wong et al., 2008**). The kinetic result implies that the DNA-binding dynamics of DPO4 may be facilitated at $\rho \leq 0.8$, where folding of DPO4 to the stable apo form is slowed down.

The folding order of elements in a protein dictates the kinetic folding pathway. Experimental determination of the domain folding order in DPO4 is challenging (**Sherrer et al., 2012**). To measure the folding order of the domains/interfaces and establish its connection to the interactions in DPO4, we calculate the probability of folding domain/interface $I$ in step $k$, termed the folding order probability $\mathrm{OP}_k^I$, for different values of $\rho$ (**Figure 3D–F**). We find a high chance of folding the domains prior

to forming the domain interface when ρ is very small ($\rho = 0.5$, *Figure 3D*). In particular, domain folding at $\rho = 0.5$ approximately follows a deterministic route with T → F → LF → P. Increasing ρ leads to more dispersed distributions of $\mathrm{OP}_k^I$ (*Figure 3E and F* and *Figure 3—figure supplement 1*). The degree of dispersion of the $\mathrm{OP}_k^I$ distribution is quantified by calculating its standard deviation $\sigma_{\mathrm{OP}_k}$ (*Figure 3G*). We observe a significant decrease in $\sigma_{\mathrm{OP}_k}$ with increasing ρ, in particular when $\rho \leq 0.9$. This indicates that the orders of folding and formation of the domains and interfaces become less deterministic as ρ increases, leading to more diverse folding pathways.

We also investigate the effects of temperature on folding kinetics by analyzing kinetic simulations performed at a temperature $0.96 T_f$, which is the optimal temperature for growth of *Sulfolobus solfataricus* (*Figure 2*). We find that most of the results obtained at this relatively high temperature, such as the 'U-shaped' ρ-dependent folding time and the more (less) deterministic folding order for lower (higher) ρ, are similar to those at room temperature $T_r$. Interestingly, we observe a plateau in MFPT for the range of ρ from 1.2 to 1.5. This is probably due to the complex ρ-dependent behaviors of intra- and interdomain folding kinetics when ρ is high, although the optimum value of ρ for achieving the fastest folding is still 0.8, which is less than the default parameter of the SBM. In addition, the folding order is not the same as that at $T_r$. The LF and P domains probably accomplish folding within the first two steps of DPO4 folding at $0.96 T_f$ when ρ is low (*Figure 3*). This may contribute to the elimination of backtracking when domains in DPO4 have a strong tendency to fold spontaneously (low ρ).

## Effects of interplay between intra- and interdomain interactions on the DPO4–DNA binding function

As a DNA polymerase (*Ohmori et al., 2001*), DPO4 synthesizes DNA molecules by assembling nucleotides. An essential step in the action of DPO4 is its DNA-binding process. We investigate the effects of the interactions in DPO4 on its function in terms of DNA binding. To describe DPO4–DNA binding, we construct a 'double-basin' SBM by adding to the original SBM the F–LF interdomain contacts in DPO4 and the DPO4–DNA native contacts identified in the crystal of the DPO4–DNA binary structure using a similar protocol proposed previously (*Whitford et al., 2007*; *Wang et al., 2012b*; *Chu et al., 2014*). The 'double-basin' SBM takes into account the effects of DNA binding and aims to capture the large-scale 'open-to-closed' conformational transition in DPO4 during DNA binding. Here, we assess the efficiency and effectiveness of the DPO4–DNA binding process in terms of thermodynamic stability and kinetic rates.

We use umbrella sampling techniques to calculate the free energy landscape of DPO4–DNA binding for values of ρ ranging from 0.5 to 1.5 (details are presented in Materials and methods and Appendix 1). We find an increase in binding affinity with strengthening interdomain interactions (increasing ρ), and the value at $\rho = 0.7$ (simulated $K_d = 2.24$ nM) is approximately equal to the experimental measurement of 3–10 nM (*Figure 4A*, details of $K_d$ calculation are presented in Appendix 1) (*Fiala and Suo, 2004*; *Maxwell and Suo, 2012*; *Raper et al., 2016*). This indicates that the interdomain interactions help to stabilize the DPO4–DNA complex. For all ρ, we observe a similar multistate DPO4–DNA binding process, which we divide into four binding states (*Figure 4B*): the completely unbound states (US) where DPO4 and DNA are widely separated ($d_{\mathrm{RMS}} > 10.0$ nm), the encounter complexes (EC) where DPO4 initiates DNA binding ($2.5$ nm $< d_{\mathrm{RMS}} < 10.0$ nm, with only transient native contacts formed), the intermediate binding states (IS) with $d_{\mathrm{RMS}} \sim 2.5$ nm, and the bound states (BS) with $d_{\mathrm{RMS}} \sim 0.1$ nm. The stabilities of both the BS and IS compared with the US are enhanced by increasing the strength of the interdomain interactions, as evidenced by the free energy landscapes. Besides, we find that during binding, DPO4 itself can have three different forms: apo-DPO4, DNA binary DPO4, and intermediate DPO4 (*Figure 4—figure supplement 1*). DPO4 in the last form has broken the LF interdomain interactions with an extended linker, serving as a conformational intermediate between apo-DPO4 and binary DPO4. In the IS, DPO4 can interconvert between the apo and intermediate forms. Increasing ρ leads to a higher population of the apo form (*Figure 4—figure supplement 1*). In the BS, DPO4 is in binary form, although large conformational fluctuations can be observed (*Figure 4—figure supplement 1*). The structural characteristics of DPO4 in the IS and BS indicate that a large-scale conformational transition of DPO4 from apo and intermediate forms to binary form should occur during binding from the IS to BS.

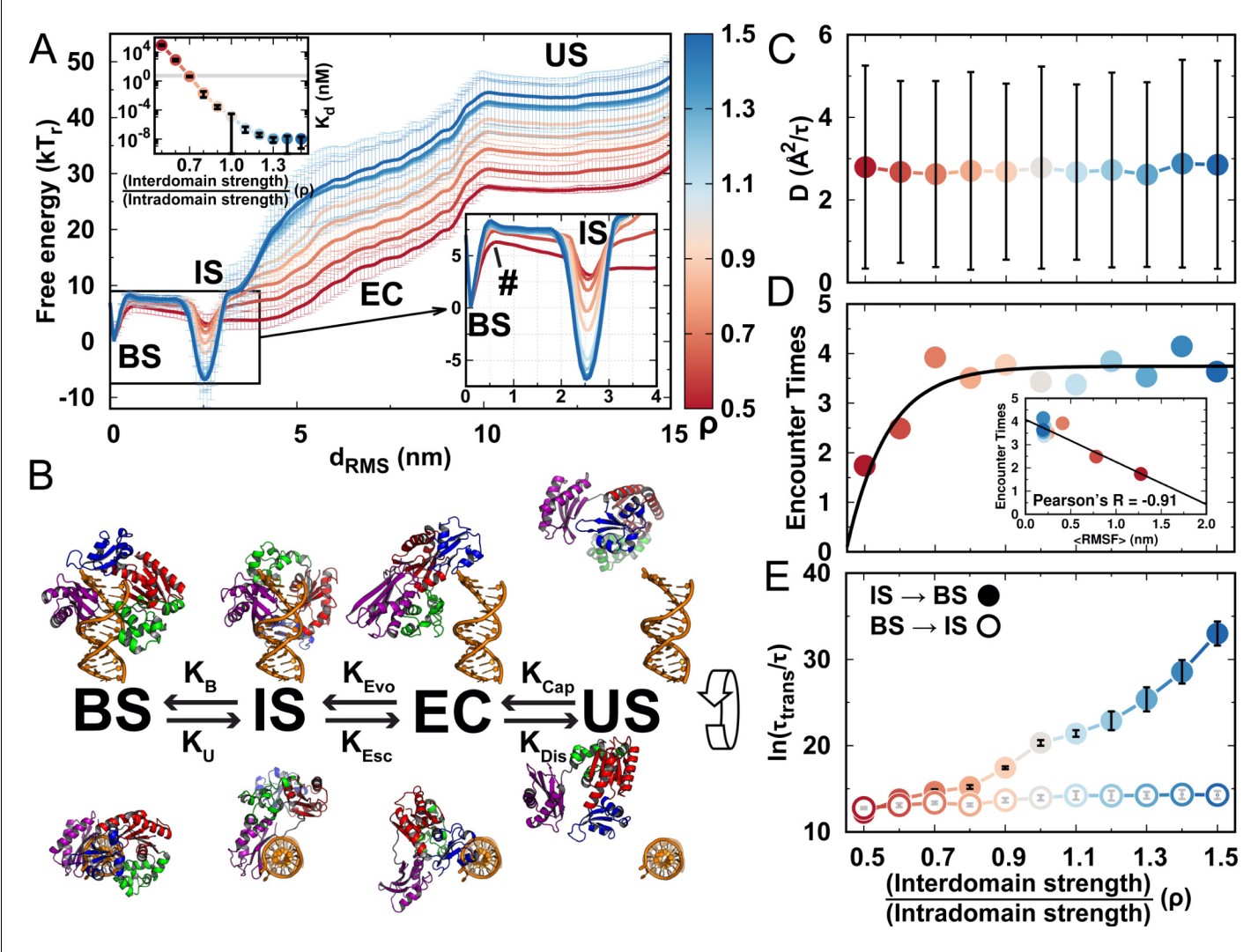

**Figure 4.** DPO4–DNA binding thermodynamics and kinetics. (A) 1D free energy landscape of DPO4–DNA binding versus the distance root-mean-square deviation of DPO4–DNA binding native contacts $d_{RMS}$ for different values of ρ. $d_{RMS}$ has units of length and deviates from 0 nm as unbinding proceeds. The binding process can be divided into four states: US, EC, IS, and BS. The insert at the top left shows the binding affinity for different values of ρ, with the gray line corresponding to the experimental measurements (3–10 nM). The insert at the bottom right is a magnified view of the binding free energy landscape focusing on the transition between the BS and the IS. The error analysis was done by performing four independent umbrella sampling simulations with different initial conditions. Error bars are omitted from this insert for clarity. (B) The four-state DPO4–DNA binding process. Typical corresponding DPO4–DNA structures obtained from the simulations for different binding states are shown from two perpendicular viewpoints. There are three transitions in the four-state DPO4–DNA binding process. The ρ-dependent DPO4–DNA binding kinetics in terms of these three transitions are shown in (C–E). (C) Diffusion coefficient $D$ of free DPO4 (in the absence of DNA) for different values of ρ. $D$ reflects the essence of $K_{Cap}$. Error bars represent the standard deviations from the corresponding mean values of $D$. τ is the reduced time unit. (D) Encounter times of the EC evolving to the IS with different values of ρ. The encounter time was calculated from the expression $k_{Dis}/k_{Evo} + 1$. The insert is a plot of encounter times versus the mean root-mean-square fluctuation (RMSF) of DPO4 in the free state. The mean RMSF was obtained from the free DPO4 simulations by averaging all the residues in DPO4 (*Figure 4—figure supplement 3*). (E) Transition times $\tau_{trans}$ between the BS and the IS for different values ρ. $\tau_{trans}$ was calculated from 100 independent simulations, and the errors were estimated by a bootstrap analysis with 50 subsamples.

The online version of this article includes the following figure supplement(s) for figure 4:

**Figure supplement 1.** Conformational distributions of DPO4 in the IS and BS.

**Figure supplement 2.** Free DPO4 diffusion for different values of ρ.

**Figure supplement 3.** Structural fluctuations of free DPO4 for different values of ρ.

**Figure supplement 4.** Thermodynamic barrier heights and kinetic times for the transitions between the IS and the BS during DPO4–DNA binding for different values of ρ.

**Figure supplement 5.** Comparison of thermodynamic and kinetic stabilities of the IS and BS during DPO4–DNA binding for different values of ρ.

To see how the interactions in DPO4 modulate the kinetics of DNA binding, we focus on the individual transitions between neighboring states during the binding process (*Figure 4B*). The transition from the US to the EC can be described approximately as a free diffusion of DPO4 to DNA molecules in the bulk. The diffusion coefficient thus controls the kinetics of this process. We perform simulations of free DPO4 in an infinite box for different values of ρ and then calculate the corresponding diffusion coefficient $D$ (*Figure 4C* and *Figure 4—figure supplement 2*). Although for low ρ, DPO4 exhibits excessive conformational flexibility (*Figure 4—figure supplement 3*), which is assumed to increase the hydrodynamic radius of the chain (*Huang and Liu, 2009*), we find little effect of the structural fluctuations on $D$. Our results indicate that the free spatial diffusion of a folded large multidomain protein is barely affected by interfacial domain formation. Therefore, the capture rate of DNA by DPO4 in the first step of the binding process should be similar for different interaction strengths of DPO4, leading to ρ-independent observations.

Spatial proximity between DPO4 and DNA does not guarantee a successful transition from the EC to the IS, since DPO4 can dissociate from DNA through thermodynamic fluctuations. To investigate how the interactions of DPO4 influence the transition from the EC to the IS, we perform 200 independent kinetic simulations for different values of ρ. Each simulation starts from random configurations of DPO4 and DNA in the EC and ends when DPO4 binds with the DNA as the IS or when DPO4 completely dissociates from the DNA as the US. We calculate the rates associated with the EC, $K_{Evo}$ and $K_{Dis}$, using a kinetic framework proposed previously (*Huang and Liu, 2009*). The encounter time, which is defined as $K_{Dis}/K_{Evo} + 1$, measures the time for DPO4 to achieve one successful transition from the EC to the IS. We find that the encounter time increases significantly as ρ increases from 0.5 to 0.7 and then becomes more or less constant for $\rho > 0.7$ (*Figure 4D*). This indicates that the weakly formed and flexible domain interface of DPO4 facilitates the DNA-binding process. We further find a strong correlation between the degree of conformational fluctuation of DPO4, quantified by the root-mean-square fluctuation RMSF, and DNA-binding encounter times. These results confirm the roles of domain interfacial fluctuations of DPO4 led by low ρ in promoting DNA binding.

Direct simulations of the transitions between the IS and the BS are expected to be computationally demanding owing to the high barriers between these two states. Instead, we calculate the kinetic rates by performing so-called frequency-adaptive metadynamics simulations (*Wang et al., 2018*), for which the computational expense is significantly lower (details are presented in Materials and methods and Appendix 1). We find that the transition times calculated from the metadynamics simulations are strongly correlated with the barrier heights measured by the umbrella sampling simulations for the transitions between the IS and the BS (*Figure 4—figure supplements 4* and *5*). The consistency between thermodynamics and kinetics resonates with the findings of previous work, where a quantitative relationship between the barrier heights and binding rates in both ordered and disordered protein-binding processes has been established (*Cao et al., 2016*). We find a monotonic increase in the kinetic times for both of the transitions between the IS and the BS as ρ increases (*Figure 4E*). In particular, significant deceleration of the transition from the IS to the BS is found as the interdomain interactions in DPO4 become stronger. By contrast, the effect on the transition rates from the BS to the IS led by ρ appear to be minor. We find a strong ρ-dependent conformational distribution of DPO4 in the IS, where DPO4 in apo and intermediate forms is dominant for high and low ρ, respectively. The population of DPO4 in the apo form in the IS hinders the conformational dynamics of transformation of DPO4 to the DNA binary form and thus is disfavored in the binding transition to the BS. On the other hand, DPO4 is almost entirely in the DNA binary form in the BS (*Figure 4—figure supplement 1*). Escape from the BS should have little dependence on the conformational dynamics of DPO4. Therefore, the transition rates between the IS and the BS are dependent on DPO4 conformational dynamics, which is modulated by its inherent interactions. It is worth noting that when $\rho \geq 0.8$, the IS state, rather than the BS state, becomes the most stable in DPO4–DNA binding (*Figure 4—figure supplement 4*), and the transition time from the IS to the BS is much longer than that from the BS to the IS. Although there are quantitative discrepancies between the thermodynamic and kinetic results (*Figure 4—figure supplement 5*), our results lead to the conclusion that to achieve and maintain the conformation that underpins its biological function, DPO4 has to avoid strong interdomain interactions, even when they are purely native.

## Discussion

Thermal denaturation experiments revealed that unfolding of truncated DPO4 mutants, such as the DPO4 core and LF domains, proceeds via cooperative processes (*Sherrer et al., 2009*). Stopped-flow Förster resonance energy transfer (FRET) studies monitoring intradomain (*Raper and Suo, 2016*) and interdomain (*Xu et al., 2009*; *Maxwell et al., 2014*) conformational dynamics of DPO4 during DNA binding as well as nucleotide binding and incorporation revealed weak and strong interactions between and within each domain of DPO4, respectively. Using only topological information, an SBM with homogeneously weighted native contacts predicted a 'divide-and-conquer' domain-wise folding scenario for DPO4 folding (*Wang et al., 2012a*; *Chu et al., 2020*). These results suggested that domains in DPO4 can fold without any aid from other domains. Previous studies showed that many multidomain proteins have stable domains that can fold independently (*Scott et al., 2002*; *Steward et al., 2002*; *Robertsson et al., 2005*). From a structural perspective, these proteins exhibit a lack of densely packed domain interfaces, so their interdomain interactions should be minimal (*Han et al., 2007*). Nevertheless, the relatively weak interdomain interactions may still play an important role in the multidomain protein folding process, since many proteins with independently folding domains are found not to fold in a 'sum of the parts' manner (*Levy, 2017*). Here, we have investigated the effects of the interplay between intra- and interdomain interactions in DPO4 on the thermodynamics and kinetics of folding. The incorporation of strength heterogeneity into the intra- and interdomain contact interactions in an SBM has enabled a quantitative investigation into how DPO4 can modulate its inherent interactions to maximize folding efficiency.

We have characterized the critical role of interdomain interactions in controlling the continuum shift of the DPO4 folding mechanism in noncooperative unstable folding, fast folding, and highly cooperative 'all-or-none' folding (*Figure 5*, left panel). Folding of multidomain proteins resembles the binding of proteins (domains) to form complexes. Two extreme cases can be outlined. One is the docking of rigid domains, and the other is the concomitant folding and binding of domains, reminiscent of binding-coupled folding in intrinsically disordered proteins (IDPs) (*Sugase et al., 2007*; *Dyson and Wright, 2002*). Which of these two mechanisms is involved in multidomain protein folding should depend on the interplay between the folding tendency of domains and the binding strength between domains (*Levy et al., 2004*; *Levy et al., 2005*). For DPO4, we have found that a decrease in the interdomain interaction strength from that in the default SBM to a value of $\rho = 0.8$ can lead to the fastest DPO4 folding rate. In the presence of weak interdomain interactions, the

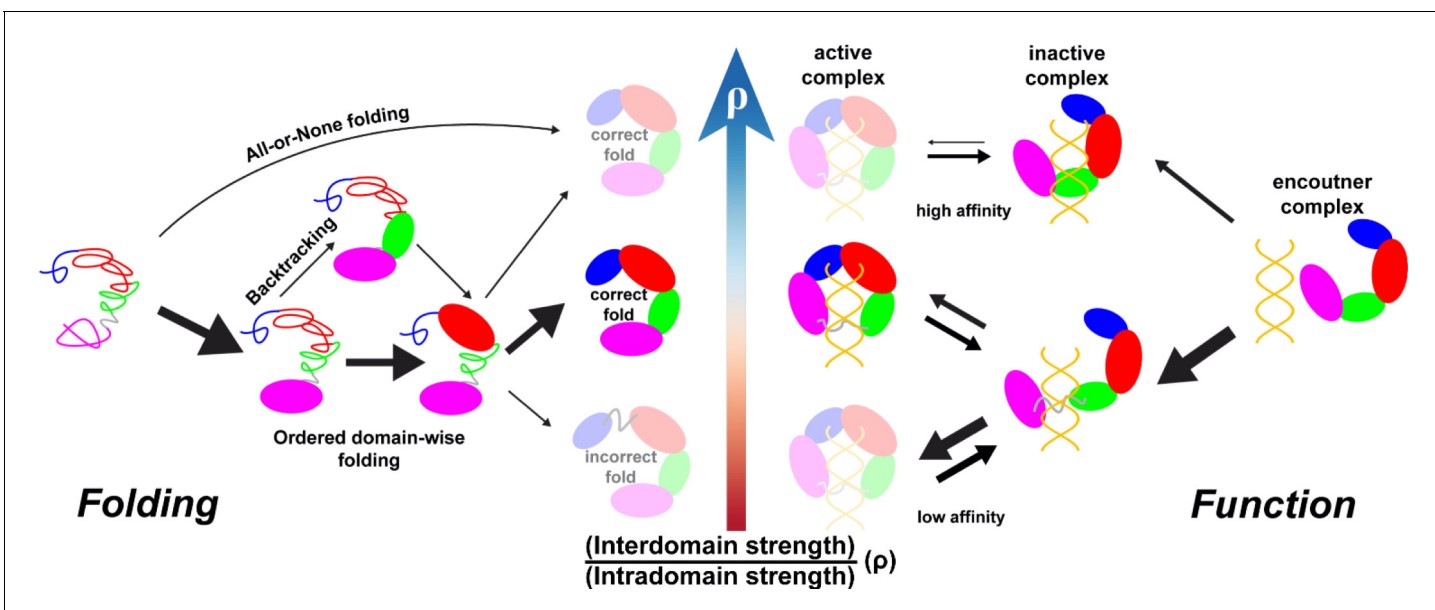

**Figure 5.** Illustration of intra- and interdomain interactions in the trade-off between DPO4 folding and function. The four domains of DPO4 are indicated using the same color scheme as in *Figure 1A*. The sizes of the arrows indicate the magnitudes of the rates or fluxes for folding (binding). DPO4 and DPO4–DNA binding complexes shown in lighter shades are less stable or populated than the others.

stability of the on-pathway folding intermediate states is enhanced. This can promote a deterministic folding order of the domains in DPO4 to help eliminate backtracking, which usually acts as a kinetic trap during folding (*Capraro et al., 2008*; *Gosavi et al., 2008*). However, if interdomain interactions are very weak, the folded states are destabilized, resulting in failed DPO4 folding. By contrast, strong interdomain interactions tend to couple domain folding and interface formation to fold the DPO4 as a whole. The 'all-or-none' DPO4 folding shows high cooperativity that disfavors the formation of intermediate states, so the folding order of DPO4 domains is not clearly defined. Besides, the domains in DPO4 in the presence of strong interdomain interactions are destabilized by the relatively weak intradomain interactions, which induces more backtracking. Collectively, the fast folding of DPO4 requires weak interdomain interactions, so that DPO4 can efficiently use the 'divide-and-conquer' strategy to fold via modest cooperativity and limited backtracking.

By investigating DPO4–DNA binding, we have addressed the roles of weak interdomain interactions of DPO4 in facilitating DNA binding. Our results show that weak interdomain interactions can induce massive conformational flexibility of DPO4 in favor of anchoring DPO4 to DNA (*Figure 5*, right panel). The result is reminiscent of the 'fly-casting' effect in the binding process (*Shoemaker et al., 2000*), where the roles of conformational disorder can be appreciated (*Huang and Liu, 2009*; *Levy et al., 2007*; *Umezawa et al., 2016*; *Chu and Muñoz, 2017*). Structural analysis has revealed that DPO4 in the IS is not in the DNA binary form (BS in *Figure 4B*) and thus is functionally inactive. At the same time, DPO4 is almost correctly located at the primer–template junction of the DNA substrate. Therefore, the transition from the IS to the BS is mainly related to the large-scale 'open-to-closed' DPO4 conformational transition. Experimentally, the existence of switching from the nonproductive to the productive DPO4–DNA complex has also been observed by single-molecule fluorescence resonant energy transfer (FRET) (*Brenlla et al., 2014*; *Raper et al., 2016*) and has been proposed to be essential in completing the DPO4–DNA binding process (*Raper et al., 2018*). Here, we have found that a decrease in interdomain interaction strength can significantly accelerate the transition from the IS to the BS, thus favoring subsequent DNA replication. On the other hand, an effect of weak interdomain interactions in facilitating the transition from the BS to the IS has also been observed, but tends to be minor. As a prototype Y-family DNA polymerase, DPO4 is capable of catalyzing translesion synthesis (TLS) across various DNA lesions (*Ohmori et al., 2001*; *Sale et al., 2012*). Weak interdomain interactions can accelerate both of the transitions between the functionally inactive IS and the active BS. Thus, they promote the function of DPO4 as a polymerase for bypassing DNA damage during TLS, which would otherwise stall DNA synthesis in vivo by high-fidelity DNA polymerases (*Kunkel and Bebenek, 2000*). It is worth noting that in reality, both very weak and very strong interdomain interactions are not favored in DPO4–DNA binding. Very weak interdomain interactions lead to high conformational flexibility in DPO4 (*Figure 4—figure supplement 3*), resulting in very-low-affinity binding of DNA, which has been widely observed in IDPs (*Wright and Dyson, 1999*; *Dunker et al., 2001*). On the other hand, very strong interdomain interactions quench the conformational dynamics of DPO4, thereby slowing the transitions during the DNA-binding process and eventually populating the binding complex at the nonproductive IS rather than the productive BS.

We see that the default SBM with homogeneously weighted intra- and interdomain interactions can lead to many consistencies in describing the DPO4 folding and DNA–binding processes with experiments. The intermediate state caused by stable individual domain folding during DPO4 folding from the default SBM simulations resonates with the experimentally observed unfolding intermediate (*Sherrer et al., 2012*). In addition, simulations with the default SBM have described a multistate process for DPO4–DNA binding, in good agreement with the results of previous experiments, where the complexity of the processes, including multistep DNA binding and multistate DPO4 conformational dynamics, was revealed (*Maxwell et al., 2014*; *Raper and Suo, 2016*; *Lee et al., 2017*). These features are manifestations that the native topologies of DPO4 and the DPO4–DNA complex are major elements in determining the mechanisms of folding (*Clementi et al., 2000*) and binding (*Levy et al., 2004*), respectively. However, there is also evidence indicating that SBM with considering only the topological factors, may not accurately capture the DPO4 conformational dynamics. The SBM simulations with various parameterizations on interdomain interaction strength show that DPO4 folding is not the most efficient when the strengths of intra- and interdomain interactions are in the same weights. Furthermore, the default SBM has led to a significantly enhanced affinity of DPO4–DNA binding and stabilized more on the functionally inactive state IS

rather than the functionally active state BS, in contradiction with the experiments. These facts imply that the default SBM fails to result in a fast-folding process of DPO4 and generate the correct functional DPO4–DNA binding. Considering both the DPO4 folding and the DPO4–DNA binding results, we have suggested that the weak interdomain interactions in DPO4 are the key to the trade-off between DPO4 folding and function. The value of ρ for achieving fast folding and efficient DNA (un) binding appears to be below 1.0. By assuming that DPO4 has structurally evolved in nature to optimize its folding and function, we suggest that the SBM with default parameter $\rho_0$ overweights the strength of interdomain interactions in its potential. This suggestion seems to be reasonable since within the domains of DPO4, there are a large proportion of conserved hydrophobic residues (*Ling et al., 2001*), which should form stronger interactions inside domains than the ones between domains. We anticipate that the SBM can be further improved by incorporating the energetic heterogeneity of native contacts (*Cho et al., 2009*) or by benchmarking available experimental observations and all-atom simulations (*Ganguly and Chen, 2011*; *Jackson et al., 2015*).

One interesting observation from our simulations is that DPO4 folds with backtracking, which has been found to be related to the fast and unstable folding followed by the unfolding of the F and T domains. Structural analysis has revealed that the F and T domains are significantly smaller than the corresponding domains in high-fidelity DNA polymerases (*Ling et al., 2001*). Thus, the interactions between these two domains and DNA are significantly reduced, providing an explanation of the ability of DPO4 to accommodate and to bypass various DNA lesions (*Ling et al., 2003*; *Vaisman et al., 2005*). Here, we have suggested that backtracking can help DPO4 to perform TLS by using the structural fluctuations of the domains. This can be undertaken by DPO4 adjusting its binding conformation through opening the active site when it encounters a bulky DNA lesion, and such an adjustment may enhance the binding of the damaged DNA to the DPO4 (*Mizukami et al., 2006*). For example, DPO4 binds DNA containing benzo[a]pyrene-deoxyguanosine and allows the bulky lesion to be flipped/looped out of the DNA helix into a structural gap between the F and LF domains (*Bauer et al., 2007*). In addition, the structure of DPO4 with DNA containing 8-(deoxyguanosine-$N^2$-yl)-1-aminopyrene (dG1,8) reveals that the dG moiety of the bulky lesion projects into the cleft between the F and LF domains of DPO4 (*Vyas et al., 2015*). These structural characteristics, differing from those of DPO4 binding to undamaged DNA, provide the dynamic basis for TLS and are probably favored by the fluctuating F and T domain conformations when binding to DNA with backtracking. However, we also note that when ρ is below 1.0 (weak interdomain interactions), the backtracking is not very populated, indicating limited fluctuations in the F and T domains. This may help to maintain the conformational rigidity of the F domain in the DPO4–DNA binding complex, which has been deemed to contribute to the low-fidelity DNA polymerization of DPO4 (*Wong et al., 2008*). Therefore, we speculate that DPO4 may optimize the extent of backtracking represented by the conformational fluctuations in the F and T domains to promote the binding of damaged DNA.

Our results have the important implication that DPO4 has naturally evolved to favor simultaneously folding and function. For folding, the high structural modularity in DPO4 has led to the high thermodynamic modularity that allows an efficient 'divide-and-conquer' mechanism by significantly reducing the number of configurations that need to be searched during folding (*Wang et al., 2012a*). Furthermore, the fastest folding of DPO4 is achieved when the interdomain interactions are weaker than they if they were determined simply by topology. This corresponds to the fact that the evolutionary pressure has acted on the DPO4 sequence to place more hydrophobic residues inside domains rather than on the domain interfaces (*Ling et al., 2001*). For DNA binding, weak interdomain interactions in DPO4 promote functional conformational transition through domain movement and facilitate all transitions throughout the DNA-binding process in favor of DNA replication and lesion bypass. Therefore, the natural evolution of DPO4 requires optimization of its protein sequence to form a structure composed of independently folded domains with hydrophobic cores to handle the folding and DNA-binding processes.

Our theoretical predictions can be potentially assessed by targeted biophysical experiments. In these, it should be straightforward to change the interactions in DPO4 via site-directed mutations and determine their effects on DPO4 folding and DNA binding. Although the positively charged linker has been targeted as one of the widely investigated mutation sites in DPO4 (*Sherrer et al., 2012*), we argue that ionic interactions can be nonspecific and long-ranged, which significantly changes the direct binding interactions with DNA (*Ling et al., 2001*). These features would lead to difficulties in delineating the effects of changes in internal interactions on the DPO4 folding and

binding processes. In this context, more attractive mutations in DPO4 would be those that disrupt the hydrophobic interactions within the domains to mimic the effect of the weakening of intradomain interactions (i.e. relatively strengthening of interdomain interactions) in theoretical studies. Subsequently, the well-developed experimental approaches for DPO4 can be used to investigate the kinetics and mechanism of DPO4 folding, DNA binding, and nucleotide binding and incorporation. For instances, the melting circular dichroism (CD) spectroscopy used in our previous study for monitoring the temperature-dependent melting of wild-type DPO4 and its various truncation mutants can be easily adapted to examine the alterations of the folding cooperativity and folding order of DPO4's domains by using site-directed mutagenesis and protein engineering (*Sherrer et al., 2012*). We expect to see a more cooperative folding of DPO4 with more synchronous melting curves of different DPO4 truncation mutants through weakening the intradomain interactions. In addition, the previously designed stopped-flow and single-molecule FRET experiments revealed a dynamical conformational equilibrium of DPO4 during DNA binding (*Xu et al., 2009*; *Maxwell et al., 2012*; *Raper and Suo, 2016*; *Raper et al., 2016*). These FRET-based techniques have further measured the kinetic rates of DPO4 interconversion between different states during DNA binding (*Raper et al., 2018*). The mutations designated to destabilize the intradomain interaction in DPO4 can potentially promote the effective interdomain interaction. Thus, according to our simulations, the conformational dynamics of DPO4 on DNA is expected to slow down because of shifting the equilibrium toward the catalytically incompetent complex. These expectations can be verified through future FRET experiments. Furthermore, the structural determination of DPO4 in complex with a damaged DNA substrate and an incoming nucleotide (*Ling et al., 2001*; *Vaisman et al., 2005*; *Ling et al., 2004b*; *Bauer et al., 2007*; *Ling et al., 2004a*; *Vyas et al., 2015*) can provide insights into the effects of backtracking caused by weakening intradomain interactions via mutations, which disrupt the structures of the ternary complexes.

Our modeling and simulations are applicable to various Y-family DNA polymerases, and we expect similar findings to be obtained for these. This expectation is based on the fact that all Y-family polymerases share structurally conserved architecture and sequence homology (*Ling et al., 2001*; *Silvian et al., 2001*; *Trincao et al., 2001*; *Uljon et al., 2004*). For polymerases in other families, the results of applying our approach may be substantially different, because these polymerases possess significantly different structural architectures to Y-family polymerases. For example, the F domains in high-fidelity DNA polymerases are much larger than those in the Y-family enzymes (*Ling et al., 2001*). A large-scale conformational change in the F domains of replicative DNA polymerases upon nucleotide binding is thought to be responsible for their high-fidelity DNA synthesis (*Swan et al., 2009*; *Johnson, 2010*; *Prindle et al., 2013*; *Bębenek and Ziuzia-Graczyk, 2018*). However, such a change is not observed with Y-family DNA polymerases. These features imply that differences in topology of multidomain polymerases lead to different folding scenarios and different biological functions. We anticipate that models with a wide range of parameters can be applied when investigating other multidomain proteins and can provide a promising way to characterize the trade-off between folding and function. Our results can offer useful guidance for protein design and engineering at the multidomain level.

# Materials and methods

## Key resources table

| Reagent type (species) or resource | Designation | Source or reference | Identifiers | Additional information |
|---|---|---|---|---|
| Software algorithm | GROMACS(version 4.5.7) | DOI:10.1002/jcc.20291 | RRID:SCR_014565 | |
| Software, algorithm | PLUMED(version 2.5.0) | DOI:10.1016/j.cpc.2013.09.018 | https://www.plumed.org/ | |
| Software, algorithm | MATLAB | MathWorks | RRID:SCR_001622 | |
| Software, algorithm | Gnuplot(version 5.2) | http://www.gnuplot.info/ | RRID:SCR_008619 | |
| Software, algorithm | Pymol(version 1.8) | Schrdödinger,Inc | RRID:SCR_000305 | |

We used an SBM to investigate the folding and DNA binding of DPO4 with molecular dynamics simulations. In SBMs, which are based on funneled energy landscape theory (**Bryngelson et al., 1995**), it is assumed that it is the native topology of the protein/complex that determines the folding (**Clementi et al., 2000**) and binding mechanisms (**Levy et al., 2004**). SBMs can be verified by comparing simulation results with experiment measurements in terms of identifying the intermediate states, $\phi$ values, folding pathways, etc. (**Clementi et al., 2000**). Here, we applied a coarse-grained SBM that used one $C_\alpha$ bead to represent one amino acid in DPO4 and three beads to represent the sugar, base, and phosphate groups of one nucleotide in DNA. For the DPO4 folding, we used a plain SBM potential $V_{\mathrm{SBM}}^{\mathrm{apoDPO4}}$ based on the crystal structure of apo-DPO4 (PDB: 2RDI) (**Wong et al., 2008**). $V_{\mathrm{SBM}}^{\mathrm{apoDPO4}}$ is made up of the bonded interactions, including bond stretching, angle bending, and dihedral rotation, as well as the nonbonded interactions (**Clementi et al., 2000**). To see the effects of intra- and interdomain native interactions on the DPO4 folding and DNA-binding processes, we further introduced a prefactor $\epsilon$ in front of the intra- and interdomain nonbonded terms to modulate the strength of the corresponding interaction. Thus, $V_{\mathrm{SBM}}^{\mathrm{apoDPO4}}$ can be expressed as follows:

$$
\begin{aligned}
V_{\mathrm{SBM}}^{\mathrm{apoDPO4}} &= V_{\mathrm{SBM}}^{\mathrm{Bonded}} + V_{\mathrm{SBM}}^{\mathrm{Nonbonded}} \\
&= V_{\mathrm{SBM}}^{\mathrm{Bonded}} + \epsilon_{\mathrm{Intra}} V_{\mathrm{SBM}}^{\mathrm{Intra}} + \epsilon_{\mathrm{Inter}} V_{\mathrm{SBM}}^{\mathrm{Inter}} + \epsilon_{\mathrm{Linker}} V_{\mathrm{SBM}}^{\mathrm{Linker}},
\end{aligned}
$$

where $V_{\mathrm{SBM}}^{\mathrm{Intra}}$, $V_{\mathrm{SBM}}^{\mathrm{Inter}}$, and $V_{\mathrm{SBM}}^{\mathrm{Linker}}$ are the nonbonded potentials for intradomain, interdomain, and linker interactions, respectively. We used a ratio $\rho = \epsilon_{\mathrm{Inter}}/\epsilon_{\mathrm{Intra}}$ to control the relative weight of the intra- and interdomain interactions. Since the linker is found to interact extensively with the other four domains in DPO4 (Table S1), we here grouped the interactions of linker into the interdomain interactions, so $\epsilon_{\mathrm{Linker}} \equiv \epsilon_{\mathrm{Inter}}$ was used throughout our simulations.

To explore DPO4 folding, we used replica-exchanged molecular dynamics (REMD) (**Sugita and Okamoto, 1999**) with the default parameters, where all prefactors $\epsilon$ are equal to 1.0 ($\rho = \rho_0 = 1.0$). Reduced units, except for the length unit (nanometers, nm), were used throughout the simulations. The weighted histogram analysis method (WHAM) (**Kumar et al., 1992**) was then applied to calculate the thermodynamics of DPO4 folding, including heat capacity curves, free energy landscapes, and domain/interface melting curves, among other things. The melting curve was further fitted by a sigmoidal function that provides the (un)folding probability along the temperature $P^I(T)$ (**Figure 2— figure supplement 6**), where $I$ indicates the individual domain/interface. $P^I(T)$ was then used to calculate TCI with the expression

$$
\mathrm{TCI}^{I,J} = -\ln \langle |[P^I(T) - P^J(T)]| \rangle,
$$

and the mean TCI (MTCI) was calculated as

$$
\mathrm{MTCI} = -\ln \left\{ \frac{1}{N^{I,J}} \sum_{I,J} \langle |[P^I(T) - P^J(T)]| \rangle \right\},
$$

where $N^{I,J}$ is the summed number of pairs $I,J$ (**Sadqi et al., 2006**). A large (small) TCI corresponds to high (low) synchronous folding of the domains/interfaces and thus a high (low) folding cooperativity of DPO4.

We used a reweighting method based on the principles of statistical mechanics to efficiently calculate the thermodynamics of DPO4 folding at other values of ρ from the REMD simulations performed at $\rho_0$ (**Cao et al., 2016**; **Li et al., 2018**). The algorithm was implemented as follows. The probability of one state having potential $E$ and reaction coordinate $r$ with parameters $\rho_0$ and ρ at temperature $T$ can be written as

$$
\begin{aligned}
p(E(\rho_0), r) &= n(r) \exp\left[ -\frac{E(\rho_0)}{kT} \right], \\
p(E(\rho), r) &= n(r) \exp\left[ -\frac{E(\rho)}{kT} \right],
\end{aligned}
$$

where $n(r)$ is the density of states. $n(r)$ is intrinsic to the system, and thus is independent of ρ (**Chu et al., 2013**). Therefore, $p(E(\rho), r)$ can be calculated by reweighting $p(E(\rho_0), r)$ as follows:

$$p(E(\rho), r) = p(E(\rho_0), r) \exp\left[-\frac{E(\rho) - E(\rho_0)}{kT}\right].$$

Since $p(E(\rho_0), r)$ has been calculated by analyzing the REMD simulations at $\rho_0$ with WHAM, we used the above equation to calculate $p(E(\rho), r)$. Eventually, the equilibrium properties for different values of ρ (free energy landscapes and averaged $Q(\text{Inter})$ along with $Q(\text{Total})$, TCI, etc.) can be obtained. The reweighting method has been proven to be effective and accurate in characterizing many other protein dynamics processes, including many-body interactions in protein folding (*Ejtehadi et al., 2004*), multidomain protein folding (*Li et al., 2018*), and protein–protein binding (*Cao et al., 2016*). These processes often require elaborate calibration of parameters in the SBM potentials, so they are always computationally expensive. To verify the results from the reweighting method, we also performed the REMD simulations for four different values of ρ (0.5, 0.8, 1.2, and 1.5) in $V_{\text{SBM}}^{\text{apoDPO4}}$. The high degree of consistency between the results of these two approaches confirms the reliability of the reweighting method (*Appendix 3—figure 2*).

To identify the backtracking and calculate the time for DPO4 folding, we performed additional kinetic simulations that ran at constant temperature. For comparing the kinetic results obtained at different ρ, it is essential that the kinetic simulations be performed under identical environmental conditions, that is, at identical temperatures. Since $T_f$ also changes with ρ, we shifted the folding temperature $T_f^\rho$ at the parameter ρ to that at $\rho_0$ ($T_f^{\rho_0}$). This was done by inserting the ratio $T_f^{\rho_0}/T_f^\rho$ in front of $V_{\text{SBM}}^{\text{apoDPO4}}$. The rationale for this is based on the fact that the solvent effects in SBMs are linearly dependent on temperature. We examined the folding temperature with the rescaled $V_{\text{SBM}}^{\text{apoDPO4}}$ for four different values of ρ (0.5, 0.8, 1.2, and 1.5) by REMD simulations and confirmed the validity of our implementation (*Appendix 3—figure 3*). Furthermore, as the DPO4 folding at $T_f$ is too slow to be represented by the kinetic simulations, we performed the simulations at two temperatures lower than $T_f$. These were the optimal temperature $T_p$ for the growth of *Sulfolobus solfataricus* and the room temperature $T_r$ of the experiments. The simulation temperatures can be approximately deduced from the linear relation

$$T_{p,r}(\text{sim}) \approx T_{p,r}(\text{exp}) \times \frac{T_f(\text{sim})}{T_f(\text{exp})},$$

where $T_f(\text{exp})$ and $T_f(\text{sim})$ are the DPO4 folding temperatures in experiments (369 K) and simulations (1.13, in energy units). $T_f(\text{sim})$ was characterized as the temperature where the heat capacity curve shows a prominent peak (*Appendix 3—figure 1*). With the experimental temperatures $T_p(\text{exp}) = 353$ K and $T_r(\text{exp}) = 300$ K, we obtain the corresponding temperatures in the simulations: $T_p(\text{sim}) = 0.96T_f(\text{sim}) = 1.08$ and $T_r(\text{sim}) = 0.81T_f(\text{sim}) = 0.92$.

For DPO4–DNA binding, we used a short DNA segment that is present in the binary DPO4–DNA PDB crystal structure (PDB: 2RDJ) (*Wong et al., 2008*). As DPO4 exhibits a large-scale 'open-to-closed' transition from apo to DNA binary structures (*Wong et al., 2008*), we built a 'double-basin' SBM for DPO4 by adding the specific contacts within DPO4 at the binary structure as the potential $V_{\text{SBM}}^{\text{binaryDPO4}}$(*Chu et al., 2014*). These contacts are found to be entirely located at the interface between the F and LF domains. It is worth noting that we did not use this 'double-basin' SBM for investigating DPO4 folding, where the potential of the SBM ($V_{\text{SBM}}^{\text{apoDPO4}}$) has only a 'single basin' in the apo-DPO4 structure. This is because (1) in the absence of DNA, DPO4 is mostly in apo form, and the transition rate for DPO4 from apo form to DNA binary form is very slow (*Raper and Suo, 2016*; *Lee et al., 2017*), so DPO4 is prone to fold to its apo form without DNA, and (2) the contacts formed at the F–LF interface can be regarded as a consequence of DNA binding (*Raper and Suo, 2016*), so the formation of these contacts reflects the effect of DNA binding. On the other hand, DNA was kept frozen throughout the simulations. Therefore, the potential for DPO4–DNA binding is expressed as

$$V_{\text{SBM}}^{\text{system}} = V_{\text{SBM}}^{\text{apoDPO4}} + V_{\text{SBM}}^{\text{binaryDPO4}} + V_{\text{SBM}}^{\text{DPO4−DNA}} + V_{\text{Elc}}^{\text{DPO4−DNA}},$$

where $V_{\text{SBM}}^{\text{binaryDPO4}}$ is the nonbonded potential (native contact) within DPO4 existing only in the DPO4–DNA binary structure, $V_{\text{SBM}}^{\text{DPO4−DNA}}$ is the native contact potential between DPO4 and DNA,

and $V_{\mathrm{Elc}}^{\mathrm{DPO4-DNA}}$ is the electrostatic potential. $V_{\mathrm{SBM}}^{\mathrm{binaryDPO4}}$ and $V_{\mathrm{SBM}}^{\mathrm{DPO4-DNA}}$ are SBM terms and provide the driving forces for the formation of the functional DPO4–DNA complex (*Levy et al., 2007*). $V_{\mathrm{Elc}}^{\mathrm{DPO4-DNA}}$ is mostly non-native, except when there is a native contact formed by two oppositely charged beads. Electrostatic interactions are known to play important roles in fast 3D diffusion (*Raper and Suo, 2016*) and in facilitated 1D search on DNA (*von Hippel and Berg, 1989*).

Only Arg and Lys in DPO4 were modeled to carry one positive charge, while and Asp and Glu in DPO4 and the phosphate pseudo-bead in DNA were modeled to carry one negative charge. In practice, we used the Debye–Hückel (DH) model, which considers the ion screening effect of a solvent to describe electrostatic interactions (*Azia and Levy, 2009*). The DH model was scaled to the strength at which two oppositely charged beads located at 0.5 nm would form the same strength as the native contact (1.0). This energy balance between the electrostatic and native contact interactions in SBM was previously suggested (*Azia and Levy, 2009*) and subsequently applied to a wide range of systems (*Chu et al., 2012*). Furthermore, we also used the rescaled $V_{\mathrm{SBM}}^{\mathrm{apoDPO4}}$ that led to the same DPO4 folding at room temperature with different values of ρ for investigating DPO4–DNA binding. Finally, we performed the DPO4–DNA binding simulations at room temperature. Details of the models can be found in SI and our previous work (*Chu et al., 2012*).

We used umbrella sampling simulations to calculate the free energy landscape of DPO4–DNA binding for different values of ρ. For a protein folding SBM, the fraction of native contacts Q is a typical reaction coordinate (*Cho et al., 2006*) and therefore is usually used for applying the biasing potential. For DPO4–DNA binding, the fraction of interchain native contacts $Q_{\mathrm{DNA}}$ is an intuitively obvious candidate for performing umbrella sampling. However, as noted elsewhere (*Cao et al., 2016*; *Chu and Wang, 2019*), $Q_{\mathrm{DNA}}$ cannot be used to distinguish among different unbound states, which all have $Q_{\mathrm{DNA}} = 0$. This would lead to great difficulty in accelerating the diffusion stage of binding and thereafter establishing the free energy for the unbound states. Instead, we chose the distance root-mean-square deviation of native contacts between DPO4 and DNA ($d_{\mathrm{RMS}}$) to perform the umbrella sampling simulations (*Chu and Wang, 2019*). $d_{\mathrm{RMS}}$ is given by

$$d_{\mathrm{RMS}} = \sqrt{\frac{1}{N} \sum_{i,j} (r_{ij} - \sigma_{ij})^2},$$

where N is the number of summed native contacts, $r_{ij}$ is the distance between pseudo-beads in DPO4 and DNA forming native contacts, and $\sigma_{ij}$ is the value of the corresponding distance at the native structure. $d_{\mathrm{RMS}}$ has a minimum value of 0 nm for the native DPO4–DNA binary structure and deviates from 0 nm as unbinding proceeds.

DPO4–DNA binding was described by the three different processes shown in *Figure 4B*. The calculations of kinetics were done by performing three additional simulations: free diffusion of DPO4, the transition from the EC to the IS, and the transition between the IS and BS. The free diffusion of DPO4 was characterized by the simulations of free DPO4 for different values of ρ. The simulations for estimating the encounter times at the EC-to-IS transition started from different EC complex configurations (defined by forming one native binding contact between DPO4 and DNA) and ran until the arrival of the IS or the complete dissociation of DPO4 from DNA.

Since a high barrier is detected between the IS and BS (*Figure 4A*), especially when ρ is high, the corresponding transitions are expected to be very slow. This makes the direct transition between the IS and BS inaccessible by the kinetic simulation. We used a kinetic rate calculation framework based on enhanced sampling simulations (*Tiwary and Parrinello, 2013*; *Salvalaglio et al., 2014*). Specifically, we applied frequency-adaptive metadynamics simulations to investigate the transition between the IS and BS (*Wang et al., 2018*). We calculated the transition times for different values of ρ and found strong correlations with the kinetics inferred from the thermodynamic free energy landscapes (*Figure 4—figure supplements 4* and *5*). The result showed the consistency between the thermodynamic and kinetic simulations (*Wang et al., 2017*; *Cao et al., 2016*). Further details of the frequency-adaptive metadynamics simulations can be found in SI Appendix 1.

## Acknowledgements

XC thank Dr. Yongqi Huang for helpful discussion of the reweighting method. The authors thank Stony Brook Research Computing and Cyberinfrastructure, and the Institute for Advanced Computational Science at Stony Brook University for access to the high-performance SeaWulf computing system, which was made possible by a $1.4M National Science Foundation grant (#1531492)

## Additional information

### Funding

| Funder | Grant reference number | Author |
|---|---|---|
| National Institute of General Medical Sciences | R01GM124177 | Zucai Suo Jin Wang |

The funders had no role in study design, data collection and interpretation, or the decision to submit the work for publication.

### Author contributions

Xiakun Chu, Conceptualization, Resources, Data curation, Software, Formal analysis, Validation, Investigation, Visualization, Methodology, Writing - original draft, Writing - review and editing; Zucai Suo, Conceptualization, Funding acquisition, Writing - original draft, Project administration, Writing - review and editing; Jin Wang, Supervision, Conceptualization, Resources, Visualization, Funding acquisition, Investigation, Writing - original draft, Project administration, Writing - review and editing

### Author ORCIDs

Xiakun Chu (ID) https://orcid.org/0000-0003-3166-7070
Zucai Suo (ID) https://orcid.org/0000-0003-3871-3420
Jin Wang (ID) https://orcid.org/0000-0002-2841-4913

### Decision letter and Author response

Decision letter https://doi.org/10.7554/eLife.60434.sa1
Author response https://doi.org/10.7554/eLife.60434.sa2

## Additional files

### Supplementary files

• Transparent reporting form

### Data availability

The necessary files for setting up Gromacs (version 4.5.7 with PLUMED version 2.5.0) simulations and analysis programs/scripts are publicly available at https://osf.io/qu5ve/.

The following dataset was generated:

| Author(s) | Year | Dataset title | Dataset URL | Database and Identifier |
|---|---|---|---|---|
| Chu X | 2020 | Folding and Binding of DPO4 | https://osf.io/qu5ve/ | Open Science Framework, qu5ve |

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

# Appendix 1

## Models and simulation protocols

The potential of SBM for DPO4 folding ($V_{\text{SBM}}^{\text{apoDNA}}$) is made up of the bonded ($V_{\text{SBM}}^{\text{Bonded}}$) and non-bonded ($V_{\text{SBM}}^{\text{Nonbonded}}$) potentials. Both are strongly biasing to the native structure of apo-DPO4 (**Wong et al., 2008**). The bonded term $V_{\text{SBM}}^{\text{Bonded}}$ has the following expression (**Clementi et al., 2000**):

$$V_{\text{SBM}}^{\text{apoDPO4}} = \sum_{\text{bonds}} K_r(r - r_0)^2 + \sum_{\text{angles}} K_\theta(\theta - \theta_0)^2 + \sum_{\text{dihedrals}} K_\phi^{(n)}[1 + cos(n \times (\phi - \phi_0))]$$

, where the parameters $K_r$, $K_\theta$ and $K_\phi$ control the strengths of bond stretching, angle bending and torsional rotation interactions, respectively. $r$, $\theta$ and $\phi$ are the bond length, angle, and dihedral angle, with a subscript zero representing the values adopted in the native structure. The non-bonded potential $V_{\text{SBM}}^{\text{Nonbonded}}$ is further subdivided into intra-, interdomain, and linker potentials based on the participating residue $i$ and $j$ (**Appendix 2—table 1**):

$$V_{\text{SBM}}^{\text{Nonbonded}} = \epsilon_{\text{Intra}} V_{\text{SBM}}^{\text{Intra}} + \epsilon_{\text{Inter}} V_{\text{SBM}}^{\text{Inter}} + \epsilon_{\text{Linker}} V_{\text{SBM}}^{\text{Linker}}$$

, where the prefactor $\epsilon$ controls the weight of each interaction term. The non-bonded potential has a Lennard-Jones-like native contact term and a purely repulsive non-native contact term, expressed as follows:

$$V_{\text{SBM}}^{(\text{Intra,Inter,Linker})} = \sum_{i<j-3}^{\text{native}} \epsilon_{ij}[5(\frac{\sigma_{ij}}{r_{ij}})^{12} - 6(\frac{\sigma_{ij}}{r_{ij}})^{10}] + \sum_{i<j-3}^{\text{non-native}} \epsilon_{PP}(\frac{\sigma_{PP}}{r_{ij}})^{12}$$

, where $\epsilon_{ij}$ and $\epsilon_{PP}$ controls the weight of native and non-native contact interactions. For native contacts, $\sigma_{ij}$ is the distance between beads $i$ and $j$ in the native structure. For non-native contacts, $\sigma_{PP}$ is equal to the diameter of the $C_\alpha$ bead and the associated interactions provide the excluded volume repulsion between the beads in DPO4. The native contact map was generated by the Contacts of Structural Unit (CSU) software (**Sobolev et al., 1999**).

Length is in the unit of nm and the others are in reduced units for all calculations, so $K_r = 10000.0$, $K_\theta = 20.0$, $K_\phi^{(1)} = 1.0$, $K_\phi^{(3)} = 0.5$, $\epsilon_{ij} = 1.0$, $\epsilon_{PP} = 1.0$ and $\sigma_{PP} = 0.4$ nm. We note in the crystal structure of apo-DPO4, there are missing residues 33–39 in the F domain. This usually implies excessive flexibility. Thus, we weakened the strength of bonded interactions within this region to 0.01 and also removed the native contacts involved by this segment. We changed the value of $\rho$, which is equal to $\epsilon_{\text{Inter}}/\epsilon_{\text{intra}}$, to modulate the interplay between the intra- and interdomain interactions. Practically, we set $\epsilon_{\text{Intra}} = 1.0$ and we changed the value of $\epsilon_{\text{Inter}} \equiv \epsilon_{\text{Linker}}$ from 0.5 to 1.5, to change $\rho$. The default parameters of SBMs have $\epsilon_{\text{Intra}} = \epsilon_{\text{Intra}} = 1.0$, so $\rho_0 = 1.0$.

The potential of SBM for DPO4–DNA binding is made up of SBM potential of apo-DPO4 ($V_{\text{SBM}}^{\text{apoDPO4}}$), SBM potentials of the specific native contact of binary DPO4–DNA ($V_{\text{SBM}}^{\text{binaryDPO4}}$ and $V_{\text{SBM}}^{\text{DPO4-DNA}}$) and electrostatic potential between DPO4 and DNA ($V_{\text{Elc}}^{\text{DPO4-DNA}}$). The electrostatic interaction was described by the Debye-Hückel (DH) model:

$$V_{\text{Elc}}^{\text{DPO4-DNA}} = K_{\text{coulomb}} B(\kappa) \frac{q_i q_j exp(-\kappa r_{ij})}{\epsilon_r r_{ij}}$$

, where $B(\kappa)$ is the salt-dependent coefficient; $q_i$ and $q_j$ are the charges of the coarse-grained beads $i$ and $j$, respectively. $\epsilon_r$ is dielectric constant and was set to 80 throughout the simulations. $\kappa^{-1}$ is the Debye screening length (in nm$^{-1}$), which is determined by the salt concentration ($C_{\text{Salt}}$, in molar units) with relation: $\kappa \approx 3.2\sqrt{C_{\text{Salt}}/C_0}$. $C_0$ is the reference molar concentration with $C_0 = 1$ M. Practically, we set $C_{\text{Salt}} = 0.15$ M and we further rescaled the strengths of the DH model, so that the two oppositely charged beads located at 0.5 nm would form the same strength of native contact (1.0). More details on electrostatic potential can be found here (**Azia and Levy, 2009**).

Thermodynamic simulations on DPO4 folding were performed at $\rho_0$ with the REMD approach (**Sugita and Okamoto, 1999**). We used 28 replicas with different temperatures ranging from 1.00 to 1.35 and concentrating around folding temperature to ensure sufficient sampling. The neighboring

replicas attempted to exchange at every 1000 MD steps following the Metropolis criterion. Each replica ran for $1 \times 10^9$ MD steps long. We observed reasonable exchanging probabilities between neighboring replicas, which are all higher than 0.2, indicating a good sampling. Finally, WHAM was used to collect all the replicas and calculate the thermodynamics of the DPO4 folding (**Kumar et al., 1992**).

Folding of individual domains of DPO4 (the F, P, T, and LF domains) was done by extracting the intradomain SBM potential from $V_{\text{SBM}}^{\text{apoDPO4}}$. For each domain, the SBM potential for folding $V_{\text{SBM}}^{\text{Domain}}$ has the bonded and non-bonded potentials that have the same expression and parameters within $V_{\text{SBM}}^{\text{apoDPO4}}$. It is worth noting that the P domain is made up of two sequentially separated segments (residues 1–10 and residues 77–166). We practically added a 5-residue long segment that has all glycines, linking the residue 10 to 77 (**Appendix 3—figure 10**). Such a long segment only has bond stretching and non-bonded repulsive potential, which has little effect on folding of the P domain. REMD simulations were performed for all the four domains independently.

Kinetic simulations for the DPO4 folding were performed for different ρ under the constant temperatures $T_p$ and $T_r$, respectively. For each ρ, we set up 100 independent simulations with a maximum length of $4 \times 10^8$ MD steps at $T_p$ and $0.2 \times 10^8$ MD steps at $T_r$, starting from different unfolded DPO4 conformations. The individual simulations were terminated when DPO4 accomplished folding, or the maximum MD steps was reached. We found that DPO4 can accomplish folding in most of the simulations (more than 91 out of 100 simulations for all ρ.). Finally, the first passage time (FPT) was collected by the criterion that the corresponding $Q$ firstly exceeds 0.75, to represent the folding rate.

We used umbrella sampling to quantify the free energy landscape of DPO4–DNA binding. $d_{\text{RMS}}$ of the DPO4–DNA binding native contact was applied as the reaction coordinate and 151 windows range from 0.0 nm to 15.0 nm were practically used. The simulation at each window was performed for $0.2 \times 10^8$ MD steps. After the first round of umbrella sampling simulation, the probability distributions of $d_{\text{RMS}}$ between 3.2 nm and 3.3 nm have little overlap when $\rho$ is large. We therefore added 10 more windows in the range of $3.2 \sim 3.3$ to ensure the sufficiency of sampling. Finally, we performed four same sets of umbrella sampling simulations (except that different initial configurations were used for different sets of umbrellas sampling). All the data were collected and analyzed by the WHAM (**Kumar et al., 1992**).

We calculated the theoretical binding affinity $K_d$ from the binding free energy landscape. We simplified the DPO4–DNA recognition to a two-state association and dissociation process:

$$[\text{DPO4}] + [\text{DNA}] \rightleftharpoons [\text{Complex}]$$

, where the 'Complex' is the (meta)stable binding complex that contains both the BS and IS. The binding affinity $K_d$ can be calculated by:

$$K_d = \frac{[\text{DPO4}] \cdot [\text{DNA}]}{[\text{Complex}]} = \frac{(1 - P_b)^2}{P_b} \cdot [\text{DPO4}]_0$$

, where $P_b$ is the probability of the binding complex and $[\text{DPO4}]_0$ is the total concentration of DPO4 in the system. In our simulation, we used a periodic cubic box with length of 20 nm, so

$$[\text{DPO4}]_0 = \frac{1 \times 10^{-3}}{N_A \times (20 \times 10^{-9})^3} mol/L = 0.21 \times 10^{-3} mol/L$$

, where $N_A$ is the Avogadro constant. $P_b$ can be calculated from the binding free energy landscape with the following expression:

$$P_b = \frac{\int_{d_{\text{rms}}=0nm}^{d_{\text{rms}}^{b/u}} exp[-F(d_{\text{RMS}})/kT_r]}{\int_{d_{\text{rms}}=0nm}^{d_{\text{rms}}=15nm} exp[-F(d_{\text{RMS}})/kT_r]}$$

, where $d_{\text{rms}}^{b/u}$ separates the binding complex and the dissociative states. In practice, different $d_{\text{rms}}^{b/u}$ can lead to different $K_d$. We calculated $K_d$ changing with $d_{\text{rms}}^{b/u}$ at different $\rho$ values (**Appendix 3—figure 12**). We found that $K_d$ does not change significantly within a range of different $d_{\text{rms}}^{b/u}$ (3.0–3.5 nm)

for separating the IS and BS at $\rho = 0.7$, when the theoretical $K_d$ approximates to the experimental $K_d$. We finally set $d_{\mathrm{rms}}^{b/u} = 3.25$ nm and obtained theoretical $K_d = 2.24$ nM, which is close to the experimental $K_d$ of 3–10 nM (*Fiala and Suo, 2004*; *Sherrer et al., 2009*).

To calculate the diffusion coefficient of free DPO4, we performed 10 independent simulations starting from folded states of DPO4 for each $\rho$. Every simulation ran for $1 \times 10^8$ MD steps. The free diffusion coefficient $D$ of DPO4 was calculated by applying the command '$g\_msd$' implemented in Gromacs (4.5.7) (*Hess et al., 2008*).

To calculate the encounter times for the transition from EC to IS, we performed 200 independent simulations starting from the DPO4–DNA configurations at EC, where only one native contact between DPO4 and DNA was formed. The simulation ran until the arrival of IS or complete dissociation of DPO4 from DNA.

To calculate the kinetics between the IS and BS, we used the frequency adaptive metadynamics approach (*Wang et al., 2018*). Metadynamics has been shown to be not only a powerful enhanced sampling method to efficiently explore the complex free energy landscape (*Laio and Parrinello, 2002*), but also a reliable approach to estimate the kinetic rate for a basin-to-basin transition (*Tiwary and Parrinello, 2013*). The acceleration factor by metadynamics was found to be:

$$\alpha(\tau_{\mathrm{sim}}) = \langle e^{V_{\mathrm{bias}}(t)/kT} \rangle$$

, where $V_{\mathrm{bias}}(t)$ is the biasing potential and the average is over a metadynamics simulation run up until the simulation time $\tau_{\mathrm{sim}}$. The kinetic calculation by metadynamics simulation is accurate if adding the bias is sufficiently infrequent so that the barrier region is not affected by the biasing potentials (*Tiwary and Parrinello, 2013*). To overcome the computational expenses laid by the infrequent bias adding, we applied an efficient approach, that is, frequency adaptive metadynamics simulation (*Wang et al., 2018*), where a strategy of the fast filling up the basin followed by the infrequent bias near the barrier was proposed.

Practically, we used $d_{\mathrm{RMS}}$ as the collective variable to add the biasing potential in metadynamics simulations. The height of the Gaussian biasing potential was set to 1.0 for the transition from BS to IS and changed to 2.0 for the transition from the IS and BS when $\rho \geq 1.0$. The well-tempered metadynamics simulation was applied with bias factors varying from 2.0 to 100.0 (increasing with $\rho$) (*Barducci et al., 2008*). The initial deposition time for adding a bias is set to 1000 MD steps and increased to the maximum of $2 \times 10^5$ MD steps, leading to the infrequent metadynamics. The details of the frequency adaptive metadynamics simulation can be found here (*Wang et al., 2018*).

We performed 100 independent frequency adaptive metadynamics simulations for the forward and backward transitions between the IS and BS for each $\rho$. A simulation for the transition from the IS to BS was deemed to a successful transition event when $d_{\mathrm{RMS}}$ is smaller than 0.1 nm, while a simulation for the transition from the BS to IS was deemed to a successful transition even when $d_{\mathrm{RMS}}$ is larger than 2.5 nm. The calculation of the transition time $\tau_{\mathrm{trans}}$ was verified using a Kolmogorov–Smirnov (KS) test (*Salvalaglio et al., 2014*) to examine whether the cumulative distribution of the transition time obeys a Poisson distribution. We found all the p-values were higher than 0.05, confirming the reliability of the calculation (*Salvalaglio et al., 2014*).

We also used free energy landscape to estimate the times for the transitions between the IS and BS based on the energy landscape theory (*Socci et al., 1996*). The calculation procedure of $\tau_{\mathrm{trans}}^*$ is described as follows. In principle, $\tau_{\mathrm{trans}}^*$ can be calculated from the double integral of the free energy landscape projected onto the reaction coordinate (here we used $d_{\mathrm{RMS}}$) with the expression (*Socci et al., 1996*; *Chahine et al., 2007*):

$$\tau_{\mathrm{trans}}^* = \int_{d_{\mathrm{RMS}}}^{d_{\mathrm{RMS}}(\mathrm{End})} d(d_{\mathrm{RMS}}) \int_{d_{\mathrm{RMS}}(\mathrm{Start})}^{d_{\mathrm{RMS}}'} d(d_{\mathrm{RMS}}') \frac{exp[(F(d_{\mathrm{RMS}}) - F(d_{\mathrm{RMS}}'))/kT_{\mathrm{r}}]}{D(d_{\mathrm{RMS}})}$$

where $D(d_{\mathrm{RMS}})$ is the position-dependent diffusion coefficient. By approximately setting $D(d_{\mathrm{RMS}})$ as a constant, we calculated the kinetic stability from the free energy landscape for different values of $\rho$. In practice, we set the thresholds of $d_{\mathrm{RMS}}$ for the transition from the IS to BS with $d_{\mathrm{RMS}}(\mathrm{Start}) = 3.5$ nm and $d_{\mathrm{RMS}}(\mathrm{End}) = 0.1$ nm and for the transition from the BS to IS with $d_{\mathrm{RMS}}(\mathrm{Start}) = 0.0$ nm and $d_{\mathrm{RMS}}(\mathrm{End}) = 2.5$ nm.

All the simulations were performed by Gromacs (4.5.7) (*Hess et al., 2008*) with PLUMED plugin (2.5.0) *Tribello et al., 2014* used for implementing umbrella sampling and metadynamics simulations. Langevin dynamics was used with a friction coefficient $1.0\tau^{-1}$, where $\tau$ is the reduced time unit. We applied constraints for all the bonds through LINCS algorithm (*Hess et al., 1997*), ensuring a $0.001\tau$ time step without any simulation instability. The non-bonded interactions were cut-off at 3.0 nm. We followed a standard protocol suggested by SMOG tools to build the input files for the SBM simulations with Gromacs (*Noel et al., 2010*).

# Appendix 2

**Appendix 2—table 1.** The native contact numbers of the intra- and interdomain, as well as the flexible Linker in apo-DPO4 PDB structure (PDB: 2RDI).

The total native contact number of the apo-DPO4 structure is 933, among which the intradomain native contact number is 787 and the number of interdomain native contacts that are mostly formed by the sequential neighbor domains, is 90. There are only two interdomain contacts formed by the non-sequential neighbor domains (P-LF interdomain interface). The number of contacts formed by the flexible Linker is 54. In DPO4–DNA binary PDB structure (PDB: 2RDJ), the T-LF contacts in apo-DPO4 PDB structure are fully broken and at the same time, 11 contacts are formed between the F and LF domains. These contacts are termed as the specific binary DPO4–DNA native contacts in constructing the 'double-basin' SBM potential ($V_{\mathrm{SBM}}^{\mathrm{binaryDPO4}}$). The other contacts remain the same in apo-DPO4 and binary DPO4–DNA structures.

|  | F domain | P domain | T domain | LF domain | Linker |
|---|---|---|---|---|---|
| F domain | 144 | 36 | 0 | 11* | 0 |
| P domain | 36 | 256 | 32 | 2 | 32 |
| T domain | 0 | 32 | 130 | 22 | 9 |
| LF domain | 11* | 2 | 22 | 257 | 11 |
| Linker | 0 | 32 | 9 | 11 | 2 |

* These 11 contacts are only formed in binary DPO4–DNA structure.

## Appendix 3

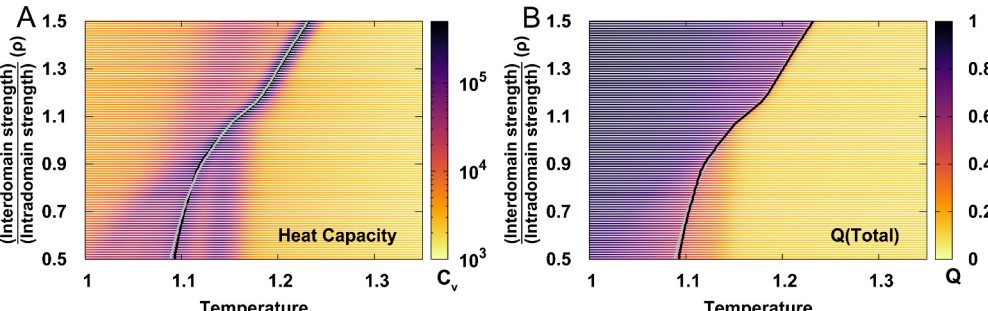

**Appendix 3—figure 1.** Extracting folding temperature $T_f^\rho$ from the heat capacity curves and melting curves of $Q(\mathrm{Total})$ for different values of $\rho$. $T_f$ can be determined as the position of the prominent peak on heat capacity curve (black lines) or the midpoint of melting curves of $Q(\mathrm{Total})$ (grey lines). These two methods generate very similar $T_f$, so $T_f$ from heat capacity was finally chosen.

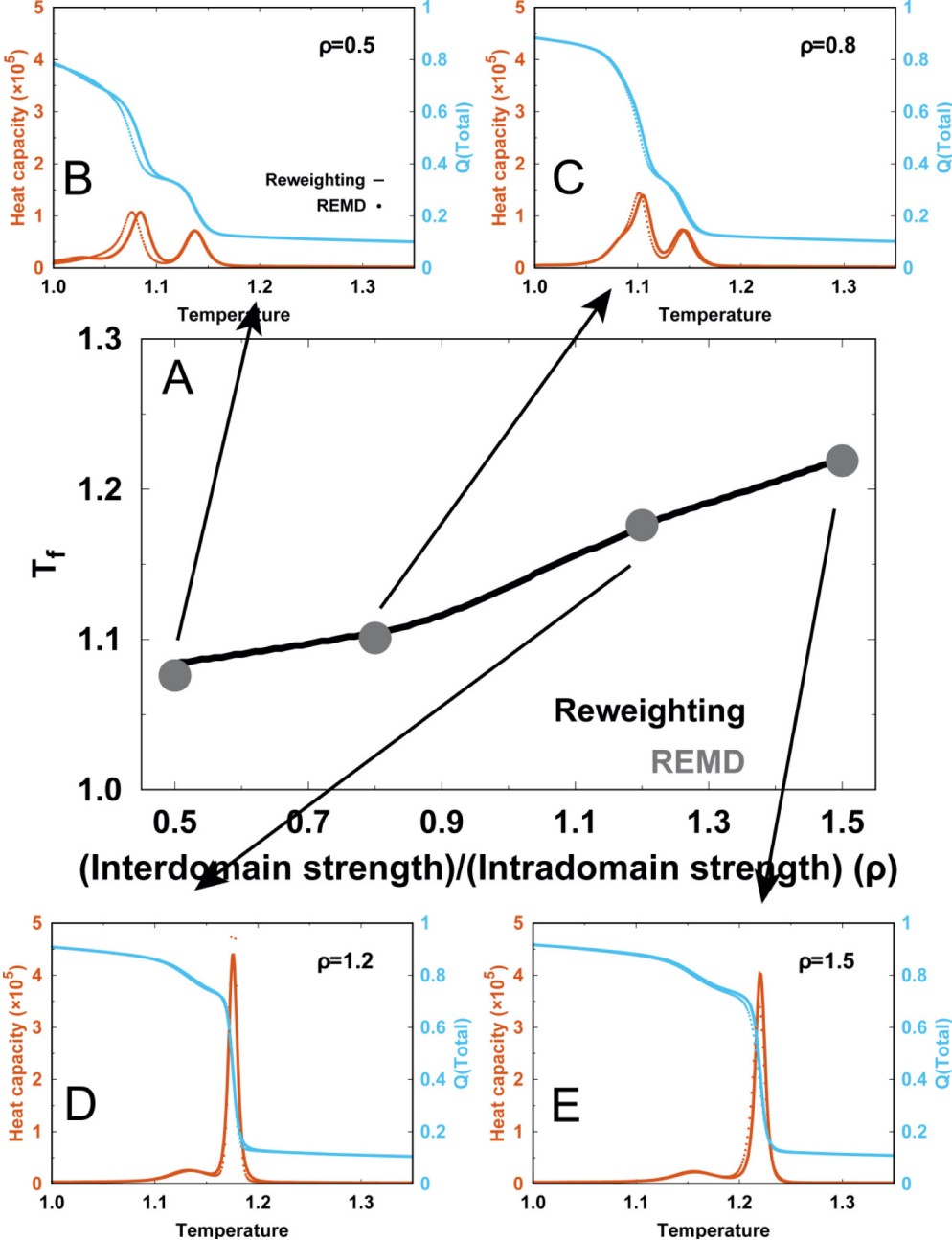

**Appendix 3—figure 2.** Comparisons of the thermodynamics between the reweighing method (solid lines) and direct REMD simulations (dots). (**A**) Comparisons of folding temperatures. The REMD simulations with directly applying four different $\rho$, which are (**B**) $\rho = 0.5$, (**C**) $\rho = 0.8$, (**D**) $\rho = 1.2$ and (**E**) $\rho = 1.5$, were performed and analyzed. In (**B-E**), comparisons of heat capacity curves (blue) and melting curves of $Q(\text{Total})$ (red) with these four different $\rho$ are shown.

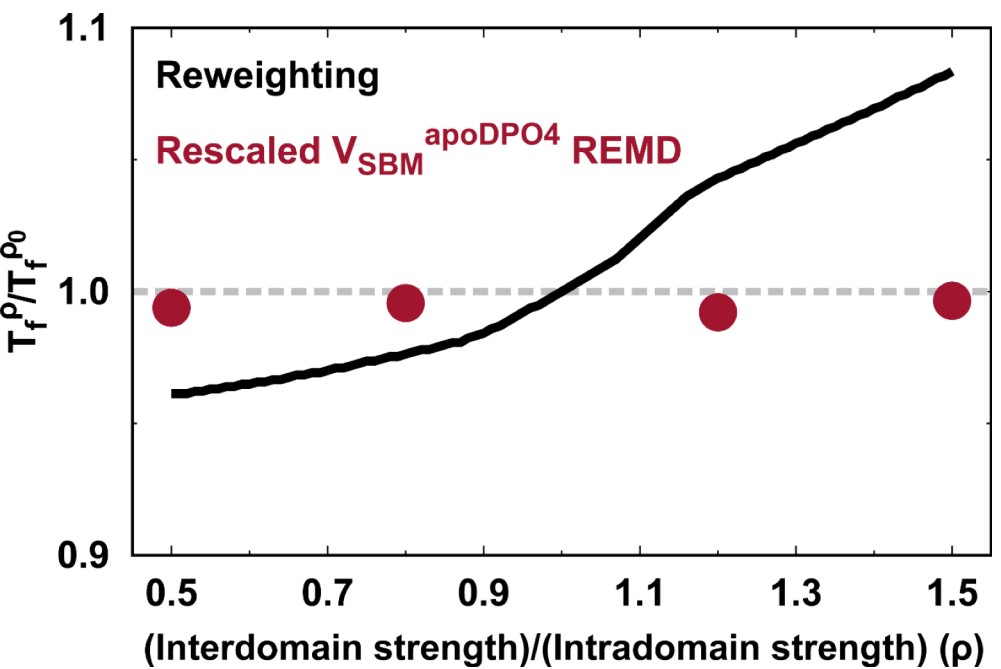

**Appendix 3—figure 3.** Assessment of shifting folding temperature with $\rho$ to that with the default parameter $\rho_0$ in order to perform kinetic simulations. Comparisons are made between the reweighting method and the direct REMD simulations with rescaled $V_{\mathrm{SBM}}^{\mathrm{apoDPO4}}$.

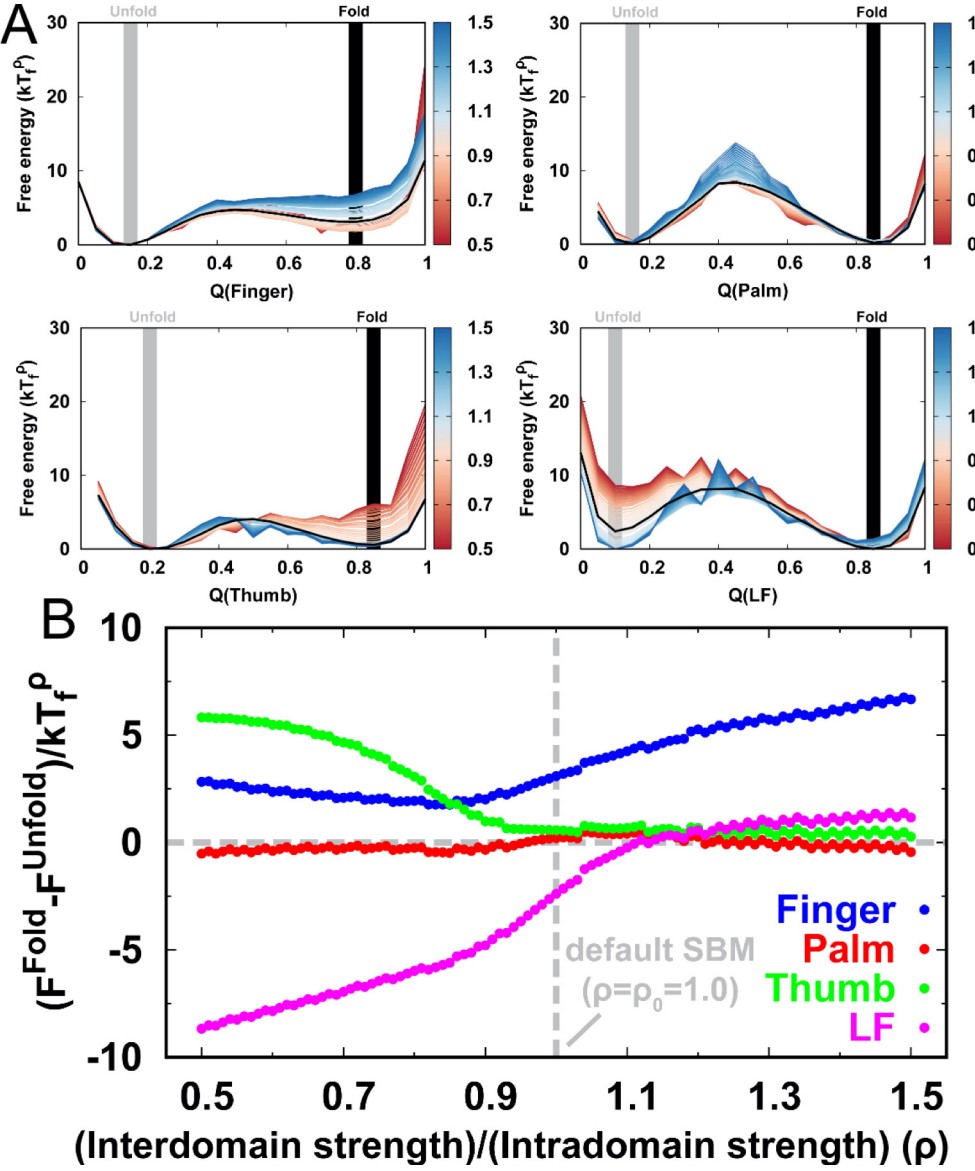

**Appendix 3—figure 4.** Folding thermodynamics of intradomain in DPO4. (**A**) Folding free energy landscapes of intradomain in DPO4 for different values of $\rho$. Temperatures are the corresponding folding temperatures varied by different $\rho$. Folded and unfolded states are marked at the $Q$ with the free energy minima. (**B**) The stability of the folded states to unfolded states varied by different $\rho$. The stability is defined as the free energy difference between the folded and unfolded states, obtained in (**A**).

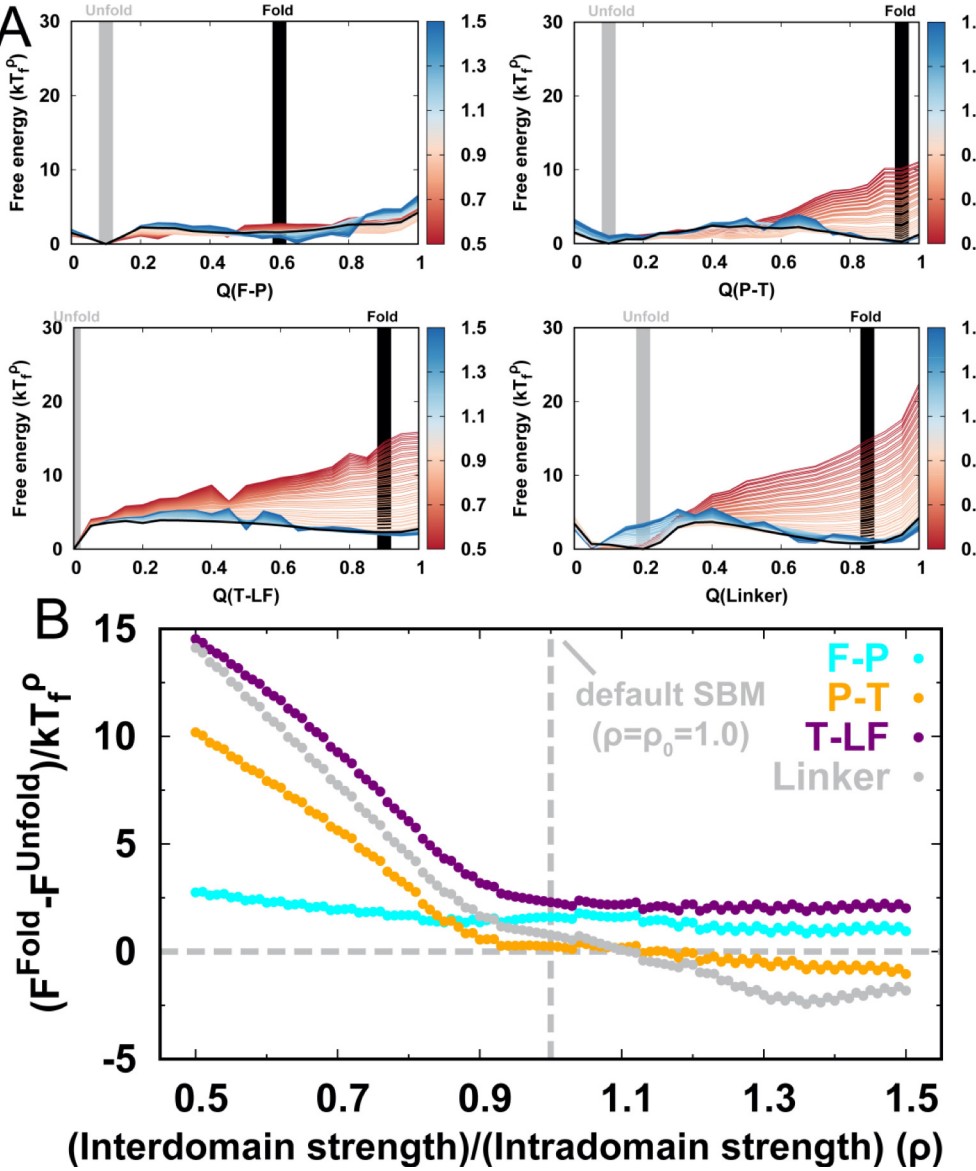

**Appendix 3—figure 5.** Folding thermodynamics of interdomain in DPO4. (**A**) Folding free energy landscapes of interdomain in DPO4 for different values of $\rho$. (**B**) The stability of the folded states to unfolded states varied by different $\rho$.

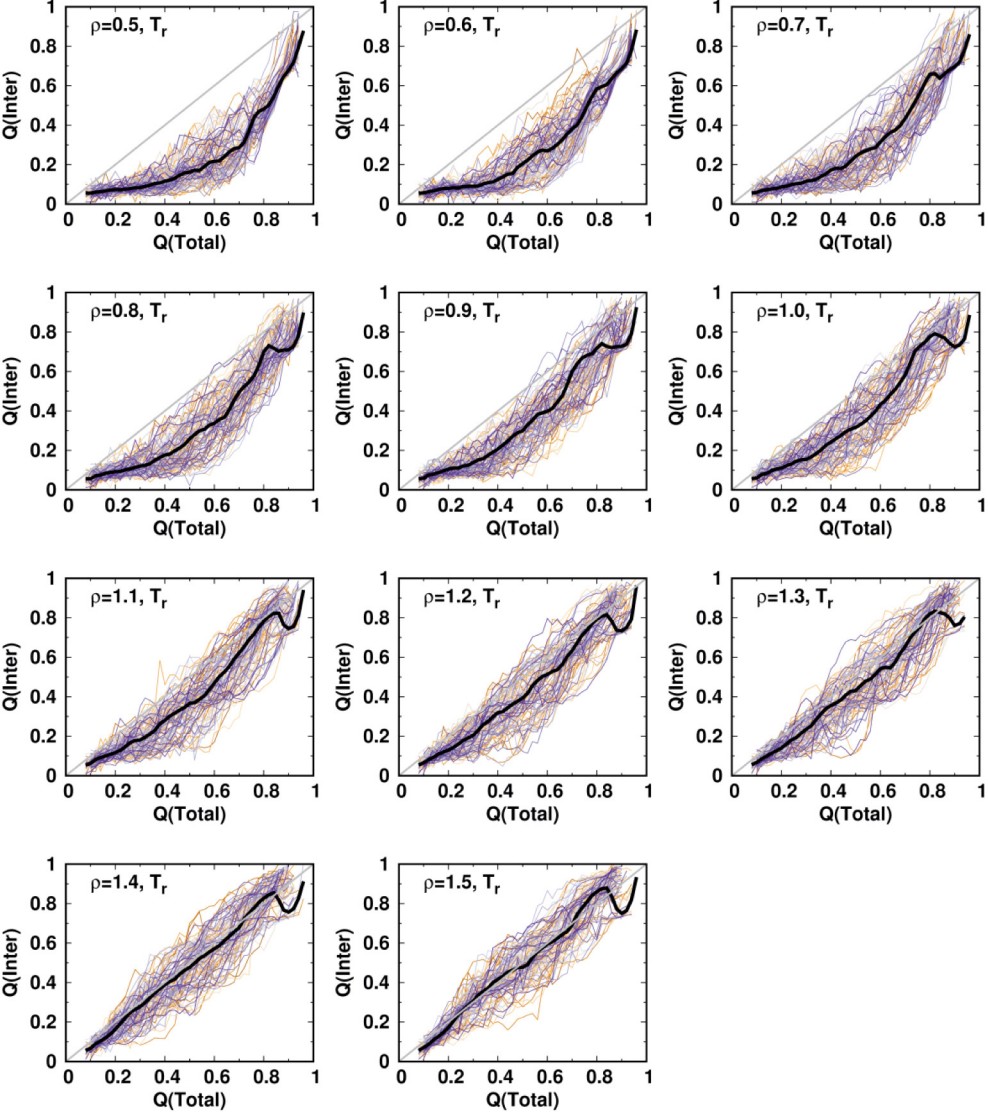

**Appendix 3—figure 6.** The averaged $Q(\mathrm{Inter})$ along with $Q(\mathrm{Total})$ at temperature $T_r$ for different values of $\rho$.

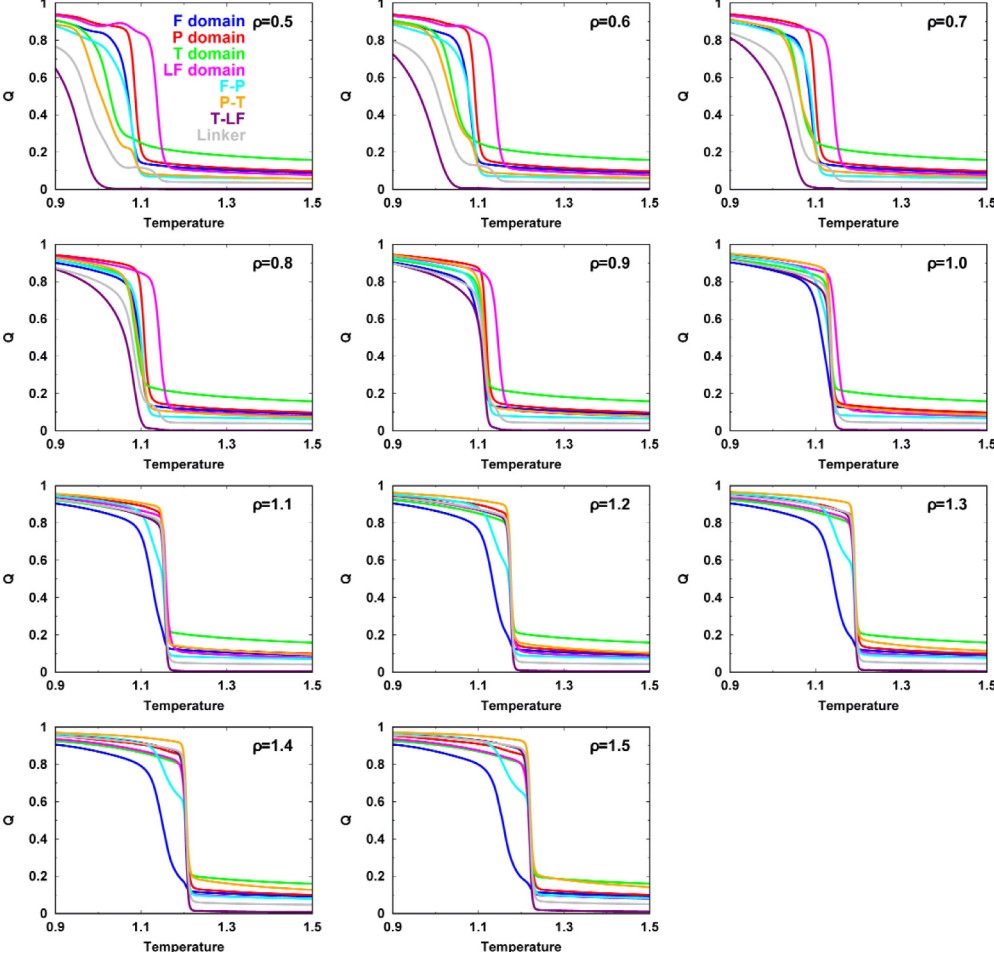

**Appendix 3—figure 7.** Melting curves of $Q$ for each intra-, interdomain of DPO4 for different values of $\rho$.

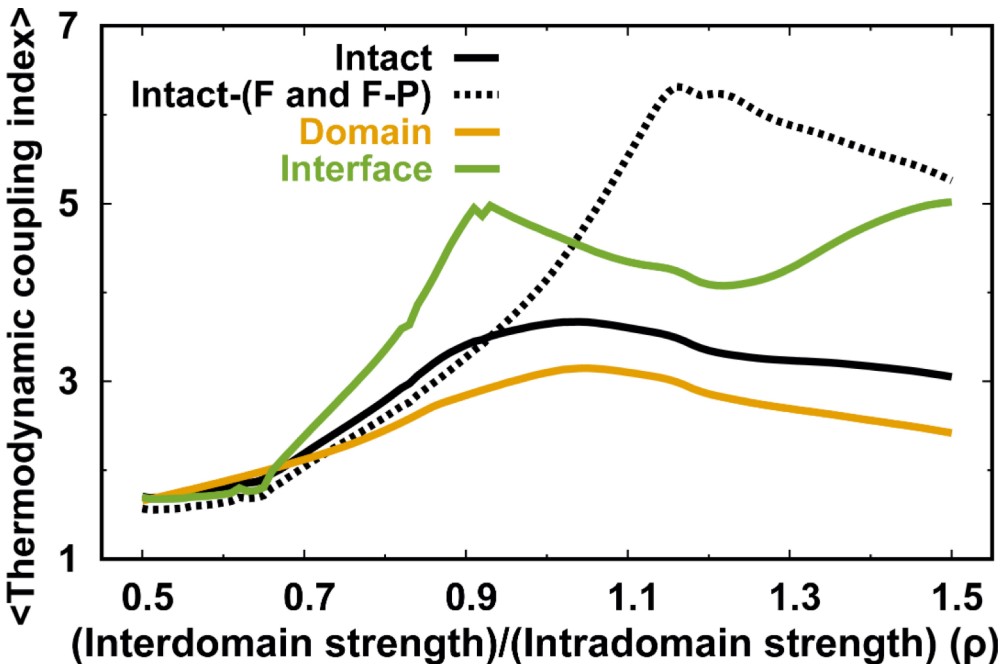

**Appendix 3—figure 8.** *MTCI* along with $\rho$. The dashed line represents the *MTCI* of removing the melting curves of the F and F-P from the intact DPO4.

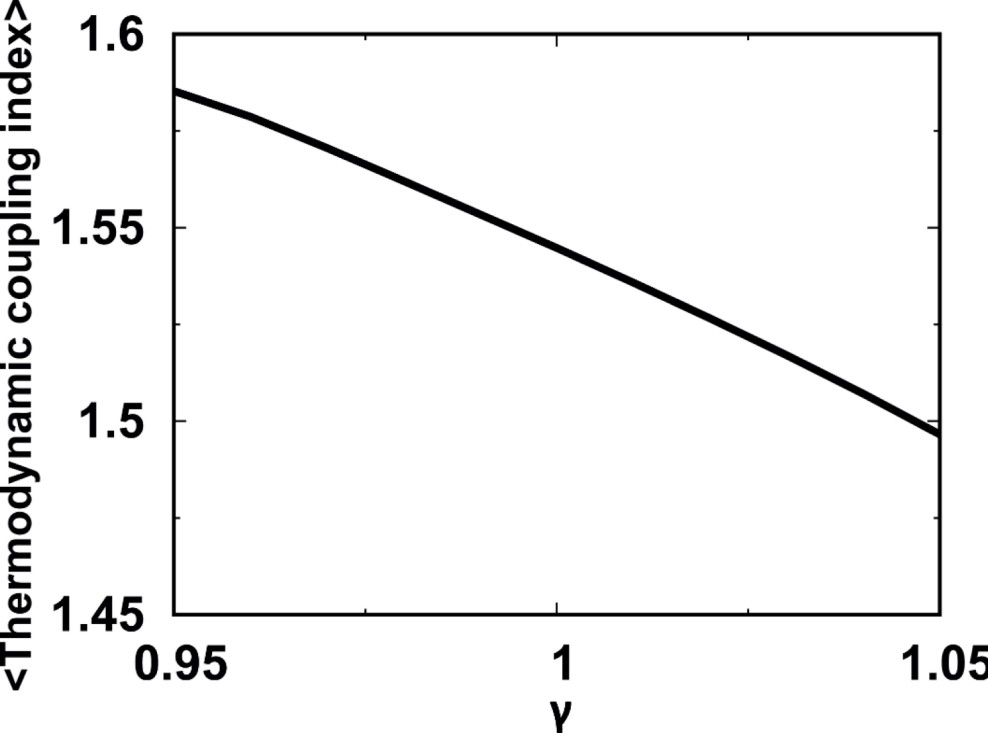

**Appendix 3—figure 9.** *MTCI* for independent folding of the four individual domains with different strengths of $V_{\mathrm{SBM}}^{\mathrm{Domain}}$ modulated by a pre-factor $\gamma$. The reweighting method is used for calculations of *MTCI* at different $\gamma$ via the REMD simulations performed at $\gamma = 1.0$.

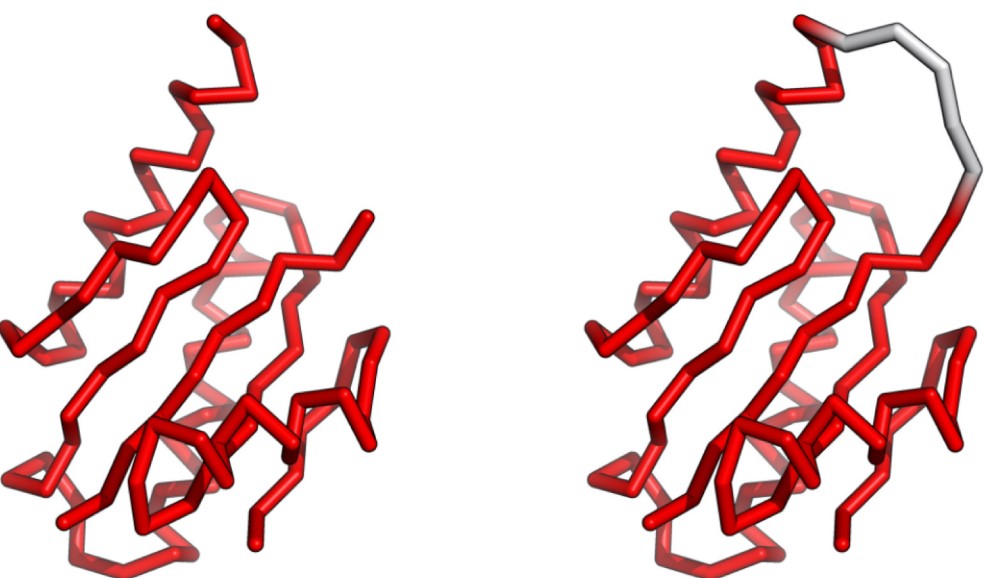

**Appendix 3—figure 10.** The coarse-grained $C_\alpha$ structures for the P domain from PDB (left) and modeling (right). Five glycines were added to connect the residue 10 to 77. The structure was used for the independent folding simulation of the P domain.

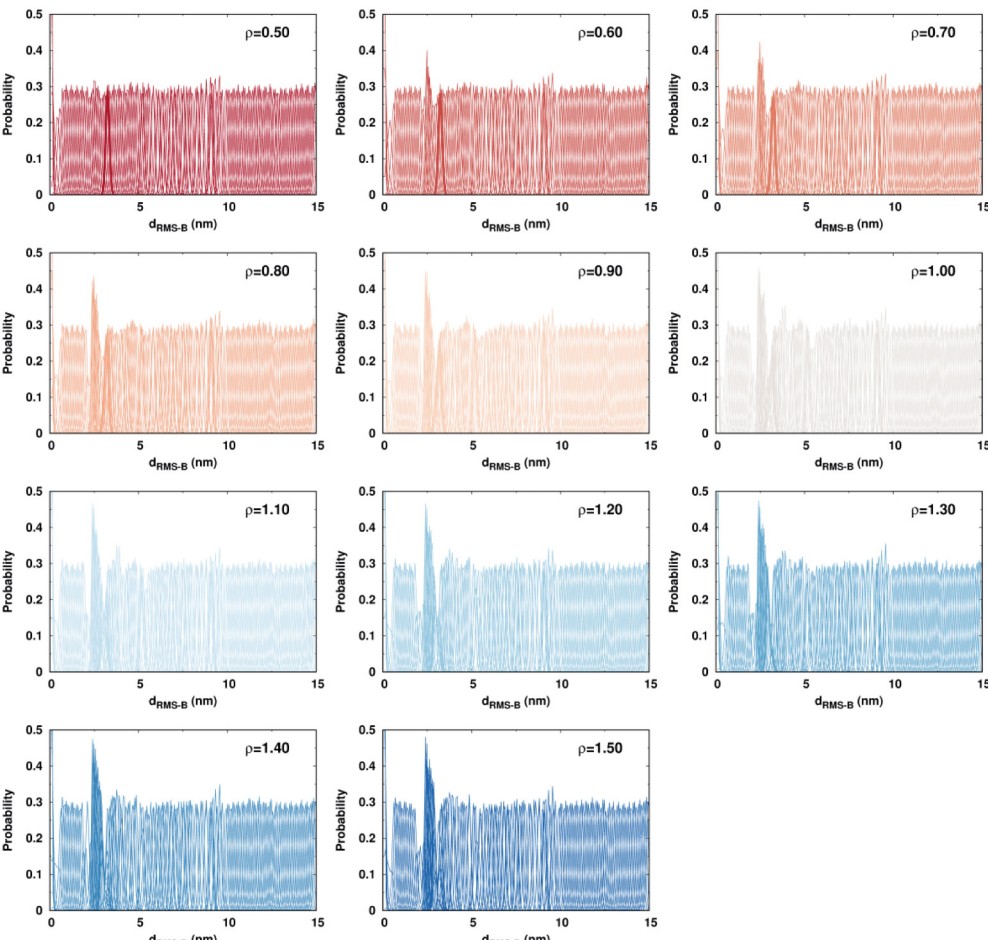

**Appendix 3—figure 11.** Biased DPO4–DNA binding simulations performed by the umbrella sampling. The overall overlaps of the probability distribution on $d_{\mathrm{RMS}}$ between neighboring replicas are apparent, indicating a sufficient sampling.

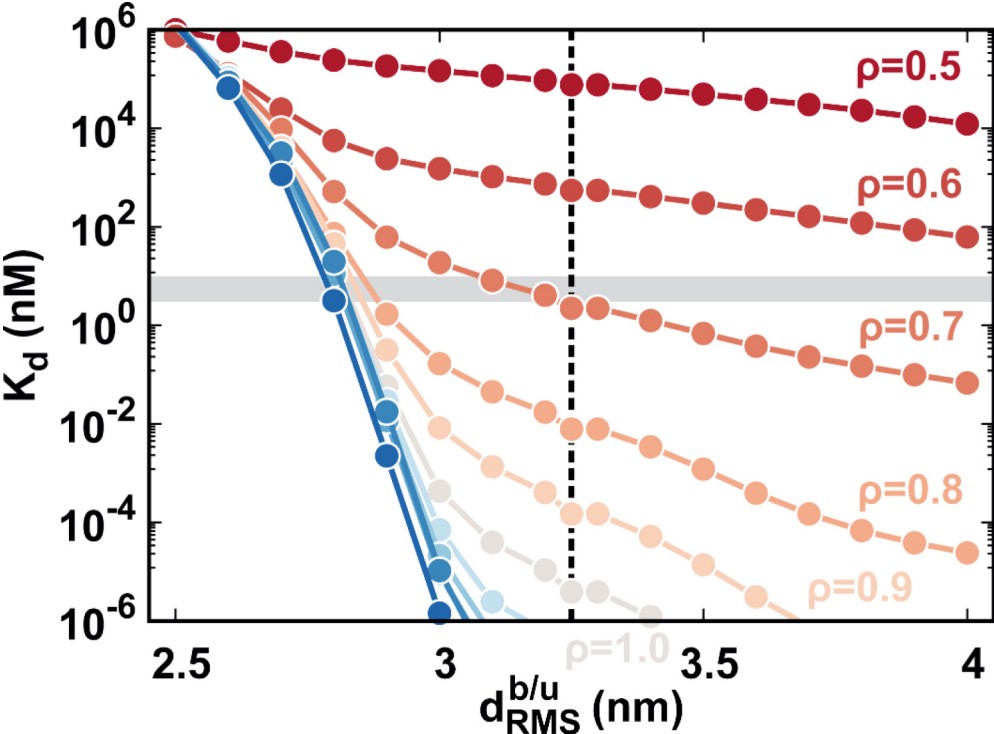

**Appendix 3—figure 12.** The calculated binding affinity $K_d$ along with $d_{\mathrm{RMS}}^{\mathrm{b/u}}$ at different values of $\rho$. The shadow region corresponds to the experimental binding affinity of 3–10 nM. The dashed line indicates the $d_{\mathrm{RMS}}^{\mathrm{b/u}}$ (3.25 nm) used in our study. The theoretical $K_d$ at $\rho$=0.7 is equal to 2.24 nM, approximating to the experimental measurements (3–10 nM).

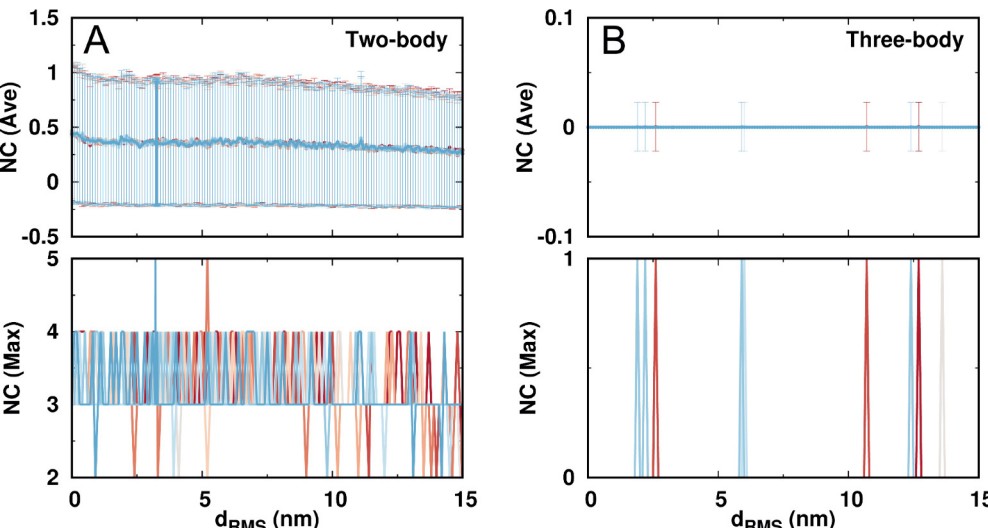

**Appendix 3—figure 13.** The number of contacts (NC) formed by the same charged beads in DPO4 during the DNA binding. The contacts are further classified into the (**A**) two-body and (**B**) three-body types. A contact is considered to form when the two beads are within the range of 0.5 nm, which is the distance for the two oppositely charged beads having DH potential strength equal to the native contact strength (1.0). The two-body (pairwise) contacts can be formed with a relatively low chance indicated by both the average and maximum contact numbers. At the same time, DPO4 is almost devoid of the three-body contacts formed by the same charged residues. This features no abnormally high number of same charges accumulated in a limited spatial space.

