## [Decision Letter]

**Acceptance summary:**

Using a Go ¯-like theoretical model Chu et al. explored the trade-off between strong intra- and inter-domain interactions of DPO4, a Y-family DNA polymerase, and showed that the system reflects a balance between expedient folding of individual domains and a stable inter-domain arrangement that also allows conformational flexibility required for the polymerase's DNA binding. The work represents an early theoretic analysis of folding of multi-domain proteins and their substrate binding.

**Decision letter after peer review:**

[Editors’ note: the authors submitted for reconsideration following the decision after peer review. What follows is the decision letter after the first round of review.]

Thank you for submitting your work entitled "From "divide-and-conquer" to "speed-stability": A trade-off between folding and function in a multi-domain protein" for consideration by *eLife*. Your article has been reviewed by three peer reviewers, one of whom is a member of our Board of Reviewing Editors, and the evaluation has been overseen by a Senior Editor. The reviewers have opted to remain anonymous.

Our decision has been reached after consultation between the reviewers. Based on these discussions and the individual reviews below, we regret to inform you that your work will not be considered further for publication in *eLife*. While recognizing the merits and sophistication of this work as, some of the reviewers are concerned that it is, as is presented in the manuscript, difficult for the more biologically oriented readership both in terms of style and in terms of substance. These reviewers also feel that publishing this work in a more biophysics-centric journal may do it more justice than *eLife*. At the same time, we would be willing to reconsider our decision if the manuscript is substantially revised to motivate the discussions by more biologically relevant questions, to make it more readable for *eLife* readership, and to give a better description of the underlying models and the simulations.

Reviewer #1:

This work used a Go model combined with enhances sampling simulations to investigate the folding of DPO4, a multi-domain protein that binds DNA. The results suggest that the relative strength of the inter- and intra-domain interactions determines the folding process, stability of the protein, and its DNA binding kinetics and affinity.

One important weakness of this manuscript, which makes this reviewer's assessment difficult, is its readability. From the main text, it is not very clear to a general reader, even one knowledgeable to protein modelling, what is the underlying physical model and what were the specific simulations. Moreover, the conclusions are presented in a somewhat convoluted way, alien to biologist. At a high level many of the conclusions are intuitive, and perhaps even obvious, e.g. strong inter-domain interactions hinder folding at domain level and leads to occasional unfolding of the domains, thus the backtracking. Overall, I am concerned whether this paper is suitable for *eLife*.

Reviewer #2:

In this manuscript, the authors use structure-based models to simulate multi-domain protein folding, in order to explore the relationship between inter- and intra-domain interactions during protein folding. Folding of multi-domain proteins is extremely challenging, and has only been seriously studied with simulations in the last few years, which makes the study timely. However, there are some serious issues with the manuscript that need to be addressed before the manuscript could be suitable for publication.

1) The use of English is rather poor. As a result, there are many passages that I cannot understand, making it difficult/impossible to assess the scientific quality of the full study. It may be necessary to consult with a professional scientific writer. Also, there are many statements for which the precision of the phrasing should be improved, such as claims of "perfect" or "proof".

2) As a motivation for the study, the manuscript states that "Until now, the mechanisms and functions of domain interactions on multi-domain protein folding have not been reported yet." However, this is not a fair statement. Specifically, the recent study of Rao and Gosavi, 2018, investigated multi-domain folding with a very similar model.

3) The authors use a re-weighting scheme to study the influence of changes in contact strengths. If this is simply free-energy perturbation, then a single equation could be given, along with a few sentence description. Since all results are derived from free-energy calculations based on re-weighting, there should be a clear description of the method, precisely as employed, in order to fully interpret the results.

4) There are many claims of backtracking, but the figures are not convincing. Specifically, Figure 2C shows that the average number of formed contacts is non-monotonic. However, this could simply arise from there being parallel pathways for folding, and the apparent backtracking could be an artifact of the projection onto *Q*. Since the simulations are based on REMD simulations, it is not obvious how one could distinguish between pathways, since full folding events are not observed. Demonstrating backtracking requires some form of pathway identification.

5) The Results state "In practice, for all different *p*, we set the free energy of the state that has the minimally formed total native contacts (*Q*(T*otal~* 0:08) to be 0." It is not clear why it is necessary to make this assumption. Couldn't the non-monotonic stability of the native ensemble be an artifact, if this assumption were not valid? As the contact strengths change, if the chosen point were to become less stable, it would shift the entire curve in Figure 1B. If the effect were sufficiently large, then perhaps the native ensemble would not exhibit the U-shape stability.

Reviewer #3:

This manuscript describes molecular dynamics simulations with the so-called Gō-like models that aim to investigate the folding and binding mechanism of a multi-domain protein modulated by inter-domain interactions. Specifically, the study focuses on determining the protein folding and DNA-binding processes of a DNA polymerase IV (DPO4). They observed an optimal DPO4 folding kinetics could be reached as inter-domain interactions were modestly weakened and showed the interesting competition between the fast, stable folding and efficient DNA-binding. This is an excellent study that is well-constructed, precisely executed and nicely presented.

Technically, they built a single-basin Gō model for DPO4 folding and double-basin Gō model for DPO4 binding with a frozen short DNA fragment. They used the classical enhanced sampling techniques, including parallel temperature replica exchanged MD and umbrella sampling, to accelerate the sampling and obtain the free energy profiles for folding and binding process, separately. They played with different strengths of inter-domain interactions 𝜖_𝐼𝑛𝑡er_ so as to modulate the ratio 𝜌 of 𝜖_𝐼𝑛𝑡𝑟𝑎_ to 𝜖_𝐼𝑛𝑡𝑟𝑎_ (the strengths of intra-domain interactions) to investigate the effects of the balance between inter-domain interactions and intra-domain interactions. To avoid extensive sampling of the free energy landscapes in different models, they employed a reweighting method to estimate the folding thermodynamics at a wide range of 𝜌 by only performing REMD simulation with the standard model (at *𝜌_0_*=1.0). I appreciated the thoroughness and the technical sophistication to solidify the strength of the results. I have some questions and a few minor suggestions that they could consider to further strengthen the work.

1) About the modeling.

Given that they aimed to explore the trade-off between folding and function for the same protein, it seems reasonable to investigate both the folding and binding process under the same energy landscape framework. Any reasons for not using a uniform double-basin Gō model for DPO4 in both folding and binding simulations?

They introduced the Debye-Hückel potential to describe the electrostatic interactions between DPO4 and DNA. It is not clear to me that if they also introduced DH potential to describe the intra-DPO4 interactions at the same time. If not, I am a bit concerned this might occur: a few positively charged residues in DPO4 bind at the same time with one negatively charged DNA bead just because these residues in DPO4 cannot feel the charges of others. Please make sure this situation didn't occur in the simulations.

And they also used specific native contacts to model the DPO4-DNA attractive interactions. So, in this binding model they used a hybrid specific LJ potential and a non-specific DH potential to model the DPO4-DNA binding process. This is of course not how real physics works in nature. Would some of the observations in this work be dependent on the choice?

There are at least four free parameters on the interaction strengths in the DPO4-DNA binding models (Materials and methods). It is not clear how the strength of the DH term was determined. And would the change of 𝜖𝐼𝑛𝑡er break the balance with other interactions? And could this impact the conclusions? Please comment on this.

2) About the simulation temperatures.

They performed kinetic simulations of DPO4 folding at the pesudo room temperature 𝑇𝑟sim which was identified by rescaling with the ratio of simulated *T_f_* to experimental *T_f_*. Changing the Hamiltonian parameters could change the thermodynamic properties, as they already recognized that "𝑇_𝑓_ also changes with 𝜌". So 𝑇𝑟sim may also change with different 𝜌. It seems the kinetic simulations were performed at the corresponding 𝑇𝑟sim recalibrated by the temperature shift caused by 𝜌 change. But it is not clear if they did the same in the DPO4-DNA binding kinetic simulations. Please clarify it.

In addition, they compared the DPO4-DNA binding affinities at different 𝜌 with experimental Kd which. But again, the simulated *T_f_* and *T_r_* may shift due to the change of 𝜌. So, does it make sense to compare the 𝜌 or T-dependent affinities with the experimental Kd which was measured at a fixed temperature?

3) They stated that "Since direct simulations on the transition between the IS and BS are computationally impractical due to the high barrier between these two states, we instead tried to infer the kinetic rates from the barrier heights for different 𝜌." But they actually didn't show the inferred kinetic rates in this manuscript, but instead just show the barrier heights in Figure 4E. I understand that they used the barrier height as a proxy of the transition rate based on the Arrhenius equation with a uniform pre-exponential factor. To release the dependence on this assumption and further strengthen the work, they could consider using other enhanced sampling methods with relatively low computational cost, such as frequency-adaptive metadynamics (Wang et al., 2018) and weighted ensemble simulation (Annu Rev Biophys. 2017;46:43-57) etc., to obtain the transition rates.

4) They stated that "We found a monotonic increase of barrier height for both two transitions between the IS and BS as 𝜌 increases (Figure 4E)." Without error estimations, it is hard to judge if the trend for BS→IS barrier increase with 𝜌 is significant or just within the errors. I would strongly suggest they do error estimations and include error bars in the free energy profiles.

5) There is one experimental author involved in this manuscript, so I read this work as an experimental/simulation collaboration, in which the simulations provide valuable predictions for experimental tests and validations. Besides the comparison with experimental Kd, it will strengthen the work by more comparisons. I understand that it is always non-trivial to combine and compare experiments and simulations, but I will appreciate if they could discuss and suggest the possibilities that could be tested by further experiments.

6) Could the conclusions in this manuscript be extended for other multi-domain proteins? Or how general are the conclusions?

[Editors’ note: further revisions were suggested prior to acceptance, as described below.]

Thank you for submitting your article "Investigating the trade-off between folding and function in a multidomain Y-family DNA polymerase" for consideration by *eLife*. Your article has been reviewed by three peer reviewers, one of whom is a member of our Board of Reviewing Editors, and the evaluation has been overseen by Cynthia Wolberger as the Senior Editor.

The reviewers have discussed the reviews with one another and the Reviewing Editor has drafted this decision to help you prepare a revised submission.

Summary:

Using a Go ¯-like theoretical model Chu et al. explored the trade-off between strong intra- and inter-domain interactions of DPO4, a Y-family DNA polymerase, and showed that the system reflects a balance between expedient folding of individual domains and a stable inter-domain arrangement that also allows conformational flexibility required for the polymerase's DNA binding. The work represents an early theoretic analysis of folding of multi-domain proteins and their substrate binding.

Revisions:

In comparison to the original submission, the reviewers agree that manuscript has been substantially improved in terms of both presentation and substance. The reviewers suggested several relatively minor revisions:

Results paragraph four indicates that the ratio of inter and intra-domain contacts was varied. How this was implemented was not clear from the text. For example, one could vary one weight, or the other, or one could modulate both, while keeping some other quantity (e.g. total stabilizing energy) constant. Without this clearly defined, it is difficult fully appreciate the potential significance of any trends based on this ratio.

The authors should expand the discussion on the potential experimental consequence of their conclusions. For example, what signature might single molecule force spectroscopy observe as an implication. What other experiments can be conducted in this regard.

The manuscript repeatedly describes effects, relative to the "default" parameterization of a structure-based model. It is not clear what significance the default parameters may have. That is, are the default parameters considered to be an accurate approximation to systems in the cell? If so, how, and to what extent? Perhaps the initial parameterization is far from appropriate for the current application, in which case variations in the ratio of different interactions may correspond to a regime that is not biologically relevant. It is important that the authors present these trends in terms of the physical insights into a biological process.

The authors may further revise their figures and make them even more intuitive. For example, rho could be labelled as relative strength of inter vs. interactions on the figure, so could MTCI. The labelling of Figure 2G is not very clear, and similar issues exist for some other figure panels.

---

## [Author Response]

[Editors’ note: the authors resubmitted a revised version of the paper for consideration. What follows is the authors’ response to the first round of review.]

While recognizing the merits and sophistication of this work as, some of the reviewers are concerned that it is, as is presented in the manuscript, difficult for the more biologically oriented readership both in terms of style and in terms of substance. These reviewers also feel that publishing this work in a more biophysics-centric journal may do it more justice than eLife.

We agree that our original manuscript was not well organized and written, as it may lead to confusion or is difficult to be understood from the biological perspective. To take this issue seriously, we have made significant changes in the manuscript by following the reviewers’ constructive critiques and useful suggestions. Furthermore, we performed additional simulations to address the reviewers’ concerns. In the end, we feel that all changes we have made in the revised manuscript are important, not only to fully respond to the critiques of yours and reviewers’, but also to make our manuscript more readable to the broad readership of *eLife*. Therefore, we sincerely thank both you and reviewers for the excellent critiques. Following is the point-to-point response to the reviewers’ comments.

Reviewer #1:[…]One important weakness of this manuscript, which makes this reviewer's assessment difficult, is its readability. From the main text, it is not very clear to a general reader, even one knowledgeable to protein modelling, what is the underlying physical model and what were the specific simulations. Moreover, the conclusions are presented in a somewhat convoluted way, alien to biologist. At a high level many of the conclusions are intuitive, and perhaps even obvious, e.g. strong inter-domain interactions hinder folding at domain level and leads to occasional unfolding of the domains, thus the backtracking. Overall, I am concerned whether this paper is suitable for eLife.

We thank reviewer #1 for pointing out the weakness of the original manuscript, which has poor readability, considering the broad readership of *eLife*. Accordingly, we have significantly revised the manuscript to address reviewer #1’s concerns. A summary of the revision made to address reviewer #1’s concern is described below: (1) A concise but informative description of the models and simulations was added in the main text for ease of reading. (2) We enriched the motivation and discussion of the model parameterization process, which now has evident connections to the biological meanings. (3) We extensively rewrote the Discussion and Conclusions section that is now easily understandable to the readers with a biological background. (4) the rewritten Introduction section includes a clear motivation of the study (this was also suggested by reviewer #2), while the Discussion and Conclusions section now has a clear understanding of the results produced by the simulations. With these four improvements, we aim to demonstrate that our study, which was rigorously performed and thoroughly compared to previous work, experimental results and even obvious intuition, has a clear aspect of novelty. We conducted a quantitative investigation through molecular simulation, for the first time, on the interplay between the intra- and interdomain interactions in modulating the folding and functional binding processes for a multidomain protein. The quantitative results have allowed us to gain an unprecedented understanding of how a multidomain protein handles the folding and substrate binding through its native topology. To summarize the central theme of our work, we have made an important discovery that a minimally frustrated landscape for folding is optimized for the functional binding process at a multidomain protein level. Our findings can be deemed as a demonstration that the action of evolutionary pressures works effectively at the multidomain protein level to produce a native structure for fast folding and efficient function. Last but not least, our theoretical modeling strategies, simulation approaches, analysis tools/methods, and conclusions can be applied to other multidomain proteins.

In conclusion, we believe that the changes we have made in our revised manuscript fully addressed reviewer #1’s concern. We hope he/she will be satisfied with the revised manuscript.

Reviewer #2:In this manuscript, the authors use structure-based models to simulate multi-domain protein folding, in order to explore the relationship between inter- and intra-domain interactions during protein folding. Folding of multi-domain proteins is extremely challenging, and has only been seriously studied with simulations in the last few years, which makes the study timely. However, there are some serious issues with the manuscript that need to be addressed before the manuscript could be suitable for publication.

As reviewer #2 indicates, one of the primary aims of our work is to explore the effects of the inter- and intra-domain interaction in a multidomain protein on the folding process. This was undertaken by the structure-based models (SBMs), which are based on the protein’s native structure and are advantageous to have an easily tunable parameterization process associated with clear experimental correspondence. We appreciate reviewer #2’s view that our work on multidomain protein folding is timely and technically feasible by the SBM-based molecular dynamics (MD) simulation. This was exactly our motivation for pursuing a theoretical and quantitative investigation of protein native structure in balancing the folding and function at a multidomain level. At the same time, the experimental approach is still limited on this aspect.

Reviewer #2 indicates that he/she has some concerns about the manuscript presentations, results, and conclusions in our original manuscript. We have taken reviewer #2’s comments very seriously and revised our manuscript accordingly. We believe that we have fully addressed reviewer #2’s concerns in the revised manuscript, which shows a more clear and solid study. We thank reviewer #2 for his/her comments and suggestions to improve our work. Our point-to-point responses are described below:

1) The use of English is rather poor. As a result, there are many passages that I cannot understand, making it difficult/impossible to assess the scientific quality of the full study. It may be necessary to consult with a professional scientific writer. Also, there are many statements for which the precision of the phrasing should be improved, such as claims of "perfect" or "proof".

We apologize for not being able to use the English properly, as it has led to difficulties for reading and hindered the understanding of our results (This was also implied by reviewer #1). Per the suggestion, we have asked a professional scientific editing agency for help in polishing and improving our manuscript. Along with the language editing service, we have also rewritten the unclear statements in our original manuscript in order to eliminate any confusion.

2) As a motivation for the study, the manuscript states that "Until now, the mechanisms and functions of domain interactions on multi-domain protein folding have not been reported yet." However, this is not a fair statement. Specifically, the recent study of Rao and Gosavi, 2018, investigated multi-domain folding with a very similar model.

We thank reviewer #2 for pointing out the study by Rao and Gosavi on investigating the folding of the serpin, a two-domain protein. By applying two SBMs that were separately built from two structures of the serpin, Rao and Gosavi found that folding of the metastable active structure is easier and faster than that of the stable latent structure. This is very inspiring and further manifests the validity of SBMs in studying the protein dynamics at the multidomain level.

While appreciating the study by Rao and Gosavi, herein we have used a theoretical modeling strategy with a parameterization of the SBM to quantitatively characterize the essential roles of the interplay between the intra- and interdomain interactions in both the folding and functional binding of DPO4. Our study, as presented in the manuscript, has shown a clear picture of changing in DPO4 folding mechanism modulated by the interdomain interactions. The DPO4 folding mechanism has been found to influence the folding kinetics critically, thus determining the folding pathways reflected by the backtracking and folding rates (This new result has been demonstrated by the additional kinetic simulations). Furthermore, the simulations on the DPO4-DNA binding process have led to an interpretation of how the DPO4 folding mechanism may influence the function (demonstrated by the thermodynamic and kinetic simulations). The detailed SBM parameterization enables us to make a quantitative connection of the DPO4 folding to its binding, thus providing a glimpse of how a multidomain protein handles the stable folding and efficient function at the same time.

In the revision, we modified the motivation of our study in the Introduction section and respectfully cited relevant work published by others. Please see the Introduction section.

“… A recent computational study of a two-domain serpin elucidated the critical role of the functional binding-related reactive center loop (RCL) in the folding of the protein to distinct structures (Rao and Gosavi, 2018). […] Addressing this issue is an important avenue in studies of multidomain protein folding.”

3) The authors use a re-weighting scheme to study the influence of changes in contact strengths. If this is simply free-energy perturbation, then a single equation could be given, along with a few sentence descriptions. Since all results are derived from free-energy calculations based on re-weighting, there should be a clear description of the method, precisely as employed, in order to fully interpret the results.

We agree with reviewer #2’s view. We have added more descriptions of the reweighting method and the explicit forms of equations that would address reviewer #2’s concern. We believe that this would also make the readers understand the technique more easily. Please see the Materials and methods section:

“We used a reweighting method based on the principles of statistical mechanics to efficiently calculate the thermodynamics of DPO4 folding at other values of *p* from the REMD simulations performed at *p_0_* (Cao et al., 2016; Li et al., 2018). […] The high degree of consistency between the results of these two approaches confirms the reliability of the reweighting method (Appendix 3—figure 2).”

4) There are many claims of backtracking, but the figures are not convincing. Specifically, Figure 2C shows that the average number of formed contacts is non-monotonic. However, this could simply arise from there being parallel pathways for folding, and the apparent backtracking could be an artifact of the projection onto Q. Since the simulations are based on REMD simulations, it is not obvious how one could distinguish between pathways, since full folding events are not observed. Demonstrating backtracking requires some form of pathway identification.

This is indeed a good comment, which has urged us to pursue a rigorous analysis of the DPO4 folding process from the kinetic perspective, in particular on the backtracking. Backtracking is defined as the process with the formation, breaking, and refolding of a subset of native contacts as the protein proceeds from the unfolded to the folded state (Gosavi et al., 2006). As indicated by reviewer #2, the rigorous identification of backtracking should rely on the kinetically undisrupted trajectories, which are not available in the REMD simulations. Therefore, we have performed additional kinetic simulations under constant temperature with different 𝜌, to verify the observation of the backtracking.

The temperature for kinetic simulations was set below the folding temperature *T_f_* of DPO4. As DPO4 folds very slowly at *T_f_*, the full folding is difficult to realize at *T_f_* during the constant temperature simulation by affordable computational efforts. With different 𝜌, we have performed 100 independent simulations at *0.96T_f_*, starting from different unfolded DPO4 configurations. Each simulation ran until one successful folding event was observed or for a maximum length of 4×10^8^𝜏, where 𝜏 is the reduced time unit. We found that DPO4 successfully folds at least in 91 out of 100 simulations with different 𝜌, leading to sufficient statistics for the pathway analysis.

We still used the same quantity, which is the averaged *Q(Inter)* along with *Q(Total)*, to describe the backtracking, as it was regarded as a standard criterion to determine the backtracking from the kinetic simulations (Gosavi et al., 2006; Gosavi et al., 2008). However, here the quantity was calculated from the individual folding trajectory rather than the combination of all the trajectories to avoid the possible mixture of the multiple folding pathways. Still, we observed an increase followed by a decrease of *Q(Inter)* along with *Q(Total)* in many individual folding trajectories during DPO4 folding as a sign of the backtracking (Figure 2D).

We further examined one trajectory and illustrated the backtracking in the DPO4 folding (Figure 2—figure supplement 3). In Figure 2—figure supplement 3, we found that the backtracking is due to the folding and unfolding of the F and T domains. These two domains during the backtracking initially fold and then unfold because of their weak folding stabilities and fast folding rates. These two domains require other domains as templates to achieve stable folding. The folding of the F and T domain can partly form the relevant interdomain interfaces, leading to an increase of *Q(Inter)*, while the subsequent unfolding of the F and T domains breaks the formation of the relevant interdomain interfaces, leading to a decrease of *Q(Inter)*. This observation in one trajectory provides a structural explanation of the backtracking detected in terms of *Q(Inter)* and *Q(Total)* (Figure 2—figure supplement 3D and E).

We have identified the backtracking in all the individual simulations through the way proposed in Figure 2—figure supplement 3C, then collected all the DPO4 structures in the backtracking from the simulations. Finally, we analyzed the intra-domain structures (Figure 2E and F). From the 1D free energy landscape projected onto *Q(Total)* (Figure 1), we can see that the backtracking mainly occurs between the *U* and *I_3_* (*I_3_’*) in Region I and the *I_3_* (*I_3_’*) and *I_2_* or *I_1_* in Region II. Along with the structural characteristics of DPO4 in the *U* (unfolded states), *I_3_* (one folded LF domain), *I_3_’* (one folded P domain), *I_2_* (folded P and LF domains) and *I_1_* (folded P, T and LF domains), we can attribute the backtracking to the transient, fast folding and unfolding of the F and T domains during the DPO4 folding.

The strength of the interdomain interaction in DPO4 has clear impacts on the backtracking during the DPO4 folding. The most probable backtracking at the region I and region II were observed with 𝜌 at 1.3 and 1.1, respectively.

Since the folding of DPO4 at *T_f_* is difficult to access, we have used a lower temperature at *0.96T_f_*. Experimentally, differing conditions can lead to different folding pathways. To see the impacts of temperatures on the backtracking during DPO4 folding, we estimated the populations of the backtracking based on the two temperatures that were used in our simulations (*0.96T_f_* and *T_r_*). We can see that from Figure 2—figure supplement 4, decreasing the temperature from *0.96T_f_* to *T_r_* (0.81*T_f_*) decreases the number of backtracking during DPO4 folding. Considering a higher temperature can lead to more instabilities of the F and T domains in DPO4, we speculate that the backtracking at *T_f_* is more significant than at the lower temperatures (*0.96T_f_* and *T_r_*).

Please see the new Figure 2 and the Results section.

“We further calculate the averaged *Q*(Inter) and *Q*(Total) and interestingly find that there are two regions exhibiting an increase followed by a decrease in *Q*(Inter) as *Q*(Total) increases (Figure 2C). […] Therefore, we suggest that the backtracking in DPO4 folding is led by unstable and fast domain folding and unfolding of the small-sized F and T domains (Figure 2—figure supplement 5).”

5) The Results state "In practice, for all different p, we set the free energy of the state that has the minimally formed total native contacts (Q(Total~ 0:08) to be 0." It is not clear why it is necessary to make this assumption. Couldn't the non-monotonic stability of the native ensemble be an artifact, if this assumption were not valid? As the contact strengths change, if the chosen point were to become less stable, it would shift the entire curve in Figure 1B. If the effect were sufficiently large, then perhaps the native ensemble would not exhibit the U-shape stability.

We made such an assumption based on the intuition that a protein with the simple SBM potential at the completely unfolded states (*Q(Total)*=0.0) has no native contact, so the free energy should be entirely contributed by the local interaction energetic term and the entropic term. As these two terms are the same with different 𝜌, so the free energy at *Q(Total)*=0.0 can be set as the same zero points for different 𝜌.

However, as noted by reviewer #2, this assumption may not be fully guaranteed, as DPO4 never reaches the point *Q(Total)*=0.0 during the simulations (the lowest value of *Q(Total)* detected in the simulations is ~0.08). Besides, the sampling at the lowest *Q(Total)* may be insufficient. This may lead to an imprecise determination of free energy at that point (*Q(Total)*~0.08).

To rigorously investigate the changes in thermodynamics led by changing the strengths of the interdomain interactions (changing r), we have used the changes of the thermodynamic stabilities of the different folding states with respect to the change from *𝜌_0_* to *𝜌* in the new analysis. The stability of the protein folding state ∆𝐹^"^ is defined as the free energy difference between the folded or folding intermediate states and unfolded states with the expression ∆𝐹^"^ = 𝐹^"^ − 𝐹^#^, where the superscript *U* indicates the unfolded states, the superscript *S* indicates any of the folding states (*I_3_*-*I_1_* and *N*) and *F* is the free energy. The folding stability, which does not require a zero energy reference point as we set in the previous analysis, also has a clear experimental correspondence, which can be obtained explicitly (e. g., by Differential Scanning Calorimetry, CD melting spectroscopy, etc.). Therefore, the change of stability at the folding state *S* led by changing 𝜌, is calculated as ∆∆𝐹(𝜌)^$^ = ∆𝐹(𝜌)^"^ − ∆𝐹(𝜌_!_)^"^. We have calculated ∆∆𝐹(𝜌)^$^ and have plotted a new figure to replace the old Figure 1C. From Figure 1C, we see that most of the conclusions in the original manuscript have been preserved (The only exception is that the relevant discussion on the unfolded states, as the stability of the unfolded state is always 0 with different 𝜌.). It is possibly due to the fact that the unfolded states of DPO4 with a large absence of non-local interactions are minorly affected by changing 𝜌.

Reviewer #3:[…]1) About the modeling.Given that they aimed to explore the trade-off between folding and function for the same protein, it seems reasonable to investigate both the folding and binding process under the same energy landscape framework. Any reasons for not using a uniform double-basin Gō model for DPO4 in both folding and binding simulations?

This is indeed a good question. Crystal structures have revealed that DPO4 is in its apo-form when DNA is not present (Wong et al., 2008). After binding to DNA, DPO4 undergoes a large-scale conformational transition that is mainly related to the rotation of the LF domain to embrace the DNA molecule associated with breaking the T-LF interface and forming the F-LF interdomain interactions (Wong et al., 2008). There are two reasons why we have used two different models for the DPO4 folding and DNA-binding processes based on the experimental evidence. (1) The crystal structures of DPO4 have shown that the isolated DPO4 is in a stable apo-form (A-form), distinct from the DNA-DPO4 binary form (B-form). As temperature increases, DPO4 was found to slightly destabilize its A-form and likely fall into a conformational equilibrium between the A-form, intermediate (I-form), and B-form (Lee et al., 2017). However, further experiments indicated that the isolated DPO4 has a small B-form population and exhibits a very low kinetic rate for the transition from the A- to B-form when DNA is not present (Raper and Suo, 2016). Moreover, there is a disordered loop in the F domain in apo-DPO4. The disorder-to-order transition on this loop can only be observed in the DPO4-DNA binary complex (Wong et al., 2008), thus, was suggested as a hallmark for the DNA binding process (Raper and Suo, 2016). The combined experimental features indicate that apo DPO4 is primarily populated in the A-form with a certain degree of conformational flexibility, while the conformational transition of DPO4 from the A-form to the B-form can only be achieved through DNA binding. (2) Building of the double-basin SBM corresponds to the addition of the F-LF interdomain native contacts from the B-form DPO4, which is induced by DNA binding, to the single-basin SBM. Regarding this, the F-LF interdomain native contacts in the double-basin SBM reflect the effects of the DNA-binding rather than of the DPO4 topology. Therefore, we consider the formation of the F-LF interface as a consequence of DNA-binding, so the relevant native contacts should be present in the double-basin SBM in DPO4-DNA binding rather than the single-basin SBM, which focuses on folding of DPO4 into the A-form without DNA binding. Please see the Materials and methods section:

“For DPO4-DNA binding, we used a short DNA segment that is present in the binary DPO4-DNA PDB crystal structure (PDB: 2RDJ) (Wong et al., 2008). […] This is because (1) in the absence of DNA, DPO4 is mostly in apo form, and the transition rate for DPO4 from apo form to DNA binary form is very slow (Raper and Suo, 2016; Lee et al., 2017), so DPO4 is prone to fold to its apo form without DNA, and (2) the contacts formed at the F-LF interface can be regarded as a consequence of DNA binding (Raper and Suo, 2016), so the formation of these contacts reflects the effect of DNA binding.”

They introduced the Debye-Hückel potential to describe the electrostatic interactions between DPO4 and DNA. It is not clear to me that if they also introduced DH potential to describe the intra-DPO4 interactions at the same time. If not, I am a bit concerned this might occur: a few positively charged residues in DPO4 bind at the same time with one negatively charged DNA bead just because these residues in DPO4 cannot feel the charges of others. Please make sure this situation didn't occur in the simulations.

In our study, we aimed to quantify the effects of the intra- and interdomain interactions on DPO4 folding. The interactions in SBMs are purely native, so they can be easily distinguished and modulated in terms of the intra- and interdomain interactions. This has enabled us to make a quantitative investigation on the effects of changing the intra- and interdomain interaction strengths in DPO4 on the folding process. As the electrostatic interactions are known to be long-ranged, and mostly non-specific/non-native, adding the electrostatic interactions to SBMs would inevitably increase the complexity of the analysis and further hinder us from drawing a definite conclusion. It is noteworthy that the previous studies using the plain SBMs without electrostatic interactions were capable of capturing many characteristics of DPO4 folding produced by the experiments (Wang et al. 2012; J. Chem. Theory Comput. (2020) 16, 2, 1319-1332). Therefore, we approximately ignored the electrostatic interaction in DPO4 folding and did not apply the Debye-Hückel (DH) model to the intra-DPO4 interactions in the simulations. On the other hand, the electrostatic interactions are crucial for the protein-DNA binding processes (performing the fast 3D diffusion close to the diffusion limit, forming the non-specific DPO4DNA binding complex and the functional DPO4-DNA binding complex, etc.), thus were included in the DPO4-DNA binding models.

Regarding the concern raised by reviewer #3, we think that reviewer #3 wants to confirm that there should not be the same charged beads in DPO4 accumulating in a limited space. Spatial accumulation of the same charged beads is unrealistic as the same charged beads would naturally form repulsive electrostatic interactions to push others away. However, the repulsive interactions between the same charged beads are missing in the intra-DPO4 model, and such an effect may be absent. To assess this situation, we have calculated the number of contacts formed by the same charged beads in DPO4 during the umbrella sampling simulations for DNA binding. In Appendix 3—figure 13, the contacts between the same charged DPO4 residues are further classified into the two-body and three-body types. We can see that the two-body (pairwise) contacts were formed with a very low chance indicated by the average and maximum contact numbers. At the same time, DPO4 was almost devoid of the three-body contacts formed by the same charged residues. This features no abnormally high number of same charges accumulating in a limited spatial space. Hence, the intraDPO4 model without electrostatic interactions is appropriate for taking care of this issue, probably due to the nature of native interactions employed in this study.

And they also used specific native contacts to model the DPO4-DNA attractive interactions. So, in this binding model they used a hybrid specific LJ potential and a non-specific DH potential to model the DPO4-DNA binding process. This is of course not how real physics works in nature. Would some of the observations in this work be dependent on the choice?

Specific native contact interactions between DPO4 and DNA are the driving forces in SBM to form the functional binding complex. SBMs by taking account into only native contact interactions have been proved very successful in describing not only the protein folding (Clementi et al., 2000) but also the protein binding (Levy, Wolynes and Onuchic, 2004) processes. The energy landscape theory claims that protein folding and binding occur on the minimally frustrated landscapes (Bryngelson et al., 1995). The non-native interactions have mainly been removed, so the native contact in principle, determines the mechanism of the molecular process. In particular, SBMs have also been used for studying protein-DNA recognition (Levy, Onuchic and Wolynes, 2007; J. Am. Chem. Soc. (2009) 131, 15084−15085) and have provided extensive predictions that are in line with the experiments (e. g., Proc. Natl. Acad. Sci. U.S.A (2011) 108, 17957-17962; Chu and Munoz, 2017). These features are the justifications of using SBMs with specific native contacts between DPO4 and DNA to investigate the binding process.

On the other hand, it is well known that electrostatic interactions are crucial in protein-DNA binding. It was found that the rate of DPO4 anchoring with the DNA is very high close to the diffusion limit, due to the facilitation made by electrostatic interactions (Raper and Suo, 2016). After forming the encounter complex, electrostatic interactions are responsible for promoting the search along with the DNA molecules in terms of 1D sliding (von Hippel and Berg, 1989). For DPO4 with a short DNA segment present, the 1D sliding has been simplified to a short-term 1D translocation process on the DNA (Chu et al., 2014). These 3D and 1D processes of DPO4 binding to DNA are controlled by the nonspecific electrostatic interactions, which should be explicitly considered and have been described by the DH model in SBM used in our study.

As mentioned by reviewer #3, there is a mixture of interactions for the oppositely charged beads that also form the native contacts. These two different types of interactions have different roles in DPO4-DNA recognition, determined by their corresponding interaction characteristics. The native contact interaction term is described by the Lennard-Jones (LJ) potential, which is short-ranged. This term is responsible for the transition from the non-specific search model to specific DNA binding complex, thus weakening or removing the native contact might fail the functional binding transition process (Chu and Munoz, 2017). On the other hand, the electrostatic interaction term is long-ranged. When DPO4 is far from the DNA native binding sites, electrostatic interaction, in principle, acts as the driving force for DPO4 to perform either the 3D diffusion or 1D translocation. Weakening or removing the electrostatic interaction formed by the native contact pairs (termed as the specific electrostatic interaction) would decrease the searching efficiency of the DPO4-DNA recognition.

Intuitively, the presence of the two different interactions formed by the oppositely charged native contact pairs seems unrealistic. In general, the specific protein-DNA binding complex is mainly stabilized by the complementary charges located at the binding interface (Proc. Natl. Acad. Sci. U.S.A (2011) 108, 17957-17962). However, the specific DNA binding site, compared to the random non-specific DNA binding sequence, should have an additional effect (cooperative interaction generated from the recognition sequence) to stabilize the specific protein-DNA binding complex as a termination of the non-specific search (Nat. Commun. (2020) 11, 540). For simplicity, we did not distinguish the specific and non-specific DNA sequences in SBM, so the electrostatic interactions from the different DNA sequences are the same. The interactions from the specific binding sites that are responsible for the binding cooperativity (the capability for switching from the non-specific to specific binding complex) have been included and described as the native contacts in SBM. However, the type of interaction is electrostatic in nature. In other words, the native contacts account for the stabilization effects originated from the specific DNA binding sequences.

Therefore, we would argue that taking into considerations both the short-ranged native contact and long-ranged electrostatic interactions are very important in building a model that is essential to describe the non-specific DNA search process and specific DNA recognition process. Please see the Materials and methods section:

“V^binaryDPO4^_SBM_ and V^DPO4-DNA^SBM are SBM terms and provide the driving forces for the formation of the functional DPO4-DNA complex (Levy et al., 2007). V^DPO4-DNA^Elc is mostly non-native, except when there is a native contact formed by two oppositely charged beads. […] Finally, we performed the DPO4-DNA binding simulations at room temperature. Details of the models can be found in SI and our previous work (Chu et al., 2012, 2014).”

There are at least four free parameters on the interaction strengths in the DPO4-DNA binding models (Materials and methods). It is not clear how the strength of the DH term was determined. And would the change of eInter break the balance with other interactions? And could this impact the conclusions? Please comment on this.

We have used the default parameters for the first three SBM potentials (*V^apoDPO4^_SBM_*, *V^binaryDPO4^_SBM_* and *V^DPO4-DNA^_SBM_*), except when the strengths of the intra- and interdomain contacts in *V_SBM_^apoDPO4^* are modulated with changing 𝜌. The default strength of the specific native contacts from the DPO4-DNA crystal structure is set to 1.0 (in reduced energy unit).

The strength of the DH term is set to be comparable to the LJ contact as suggested by Azia and Levy (J. Mol. Biol. (2009) 393, 527-542). In practice, we have rescaled the pre-factor of the DH term so that the oppositely charged pairs located at 0.5 nm have electrostatic interaction energy of -1.0, which is equal to the native contact strength. This electrostatic and contact strength balance has been widely used in the SBM simulations (e. g., Chu et al 2012; 2014; 2019).

Here, we aimed to see how the interaction of the intra-DPO4 affects the DNA binding process. We have changed the strengths of the intra- and interdomain interactions in DPO4 (*V_SBM_^apoDPO4^*), while the last three potentials, which are related to DNA binding, remain unchanged. In addition, *V_SBM_^apoDPO4^* has been further rescaled to maintain the same folding temperature of the isolated DPO4 with different 𝜌. Therefore, the overall balance between the four terms in the potential will not change. However, as mentioned by reviewer #3, for individual contacts, the change of 𝜌 will break the previous balance among the native contact within DPO4, DPO4-DNA native contact and electrostatic interaction at the default parameter (*𝜌_0_=1.0*). Nevertheless, this is exactly what we want to see: how the particular folding interactions in DPO4 affect DPO4-DNA binding (e. g. how a particular mutation in DPO4 changes the folding and also the DNA binding). We are also aware of the extreme situations where a big change of the intra- and interdomain interaction may significantly change interaction balance at the individual contact level. This will lead to a serious issue as the interaction balance between the folding and binding interaction is entirely broken, leading to an unrealistic model and then questionable results. In this study, we have applied a moderate change of 𝜌 from 0.5 to 1.5, so the balance should have been reasonably maintained at the individual contact level. Please see the Materials and methods section. This has also been included in the above response.

2) About the simulation temperatures.They performed kinetic simulations of DPO4 folding at the pesudo room temperature T_r_(sim) which was identified by rescaling with the ratio of simulated T_f_ to experimental T_f_. Changing the Hamiltonian parameters could change the thermodynamic properties, as they already recognized that "T_f_ also changes with r". So T_r_(sim) may also change with different 𝜌. It seems the kinetic simulations were performed at the corresponding T_r_(sim) recalibrated by the temperature shift caused by 𝜌 change. But it is not clear if they did the same in the DPO4-DNA binding kinetic simulations. Please clarify it. In addition, they compared the DPO4-DNA binding affinities at different with experimental Kd which. But again the simulated T_f_ and T_r_ may shift due to the change of 𝜌. So does it make sense to compare the 𝜌 or T-dependent affinities with the experimental Kd which was measured at a fixed temperature?

We appreciate reviewer #3 for the useful comments regarding the simulation temperature. It is well known that the temperature is a critical factor in SBMs, so we have paid particular attention to setting the simulation temperature in order to make our results reliable and also comparable to the experiments. In general, the temperature in the simulations with SBM, which is an energy landscape-based physical model, does not directly connect to the real one. This is different from the temperature in MD simulations using a general empirical force field (e. g., Amber, CHARMM and OPLS/AA etc.), of which the parameters are determined by the quantum mechanics or experiment measures. Nevertheless, the temperature in the SBM simulation can be approximately connected to the real one based on consistent observations with the experiments, such as the folding temperature and B-factor (Jackson et al., 2015). Here, we have used the folding temperature to establish the connections between the simulations and the experiments.

As also noted by reviewer #3, we have observed that *T_f_* changes with 𝜌. To remove the effect of 𝜌 on the temperature, we have assigned a ratio of *T_f_^𝜌0^/T_f_^𝜌^* as the pre-factor of *V_SBM_^apoDPO4^* based on the assumption that the temperature is linearly dependent on the energy in SBM. Therefore, after this implementation, the temperature scaling of SBMs with different 𝜌 will be the same. We examined the folding temperature under the rescaled *V_SBM_^apoDPO4^* at four different 𝜌, and confirmed the validity of such implementation (Appendix 3—figure 3). In the end, all the SBMs with different 𝜌 will generate the same folding temperature *T_f_*, thus have the same temperature scaling (the same room temperature *T_r_* and others). The temperature effects on the kinetics (both folding and DNA-binding) led by different 𝜌 does not exist.

In order to compare with experiments, all the folding kinetic simulations were performed at the experimental temperatures. The experimental temperature in simulations has been estimated using the linear relation:𝐓𝐫(sim)≈𝐓𝐫(exp)𝐓𝐟(𝐬𝐢𝐦)𝐭𝐟(𝐞𝐱𝐩)

Since *T_f_(sim)* are the same for different 𝜌, *T_r_(sim)* will also be the same for different 𝜌. This allowed us to perform the folding of DPO4 with different 𝜌 at the same temperature (corresponding to the same environmental condition). For DPO4-DNA binding, we have also used the rescaled *V_SBM_^apoDPO4^* in building the DPO4-DNA binding SBM. We have performed the DPO4-DNA binding simulation at the room temperature, where the experimental *K_d_* was measured. As the folding temperatures of DPO4 for different 𝜌 has been shifted to the same by the rescaled *V_SBM_^apoDPO4^*, *T_r_(sim)* are also the same for different 𝜌. This has also led to the same external environmental conditions (fixed temperature) for simulating the DPO4-DNA binding process.

Please see the Materials and methods section. We have done several modifications to make the description clear.

“To identify the backtracking and calculate the time for DPO4 folding, we performed additional kinetic simulations that ran at constant temperature. […] With the experimental temperatures *T_p_*(exp) = 353 K and *T_r_*(exp) = 300 K, we obtain the corresponding temperatures in the simulations: *T_p_*(sim)=0.96 *T_f_* (sim)=1.08 and *T*_r_(sim)=0.81 *T_f_*(sim)=0.92.”

3) They stated that "Since direct simulations on the transition between the IS and BS are computationally impractical due to the high barrier between these two states, we instead tried to infer the kinetic rates from the barrier heights for different 𝜌." But they actually didn't show the inferred kinetic rates in this manuscript, but instead just show the barrier heights in Figure 4E. I understand that they used the barrier height as a proxy of the transition rate based on the Arrhenius equation with a uniform pre-exponential factor. To release the dependence on this assumption and further strengthen the work, they could consider using other enhanced sampling methods with relatively low computational cost, such as frequency-adaptive metadynamics (Wang et al., 2018) and weighted ensemble simulation [Annu Rev Biophys. 2017;46:43-57] etc., to obtain the transition rates.

Reviewer #3 suggested us calculating the “real” kinetic rates rather than inferring the kinetic information from the thermodynamic barrier. He/She also suggested two interesting papers that are helpful to calculate the kinetics from the enhanced sampling simulations. We appreciate his/her comments, which have inspired us to study the rare-event kinetics using reasonable computation resources during the revision. As per suggestion, we have performed additional frequency adaptive metadynamics simulations for DPO4-DNA binding with different 𝜌. We focused on the transition processes between the specific BS and non-specific IS, and then calculated the kinetic times between the IS and BS through the methodologies developed previously (Tiwary and Parrinello, 2013; Salvalaglio, Tiwary and Parrinello, 2014; Wanf et al., 2018).

From Figure 4E, we see an apparent increase of transition time from the IS to BS as 𝜌 increases (interdomain interaction strengthens) and a slight increase of transition time from the BS to IS as 𝜌 increases. This observation of the 𝜌-dependent kinetic trend is similar to that inferred from the thermodynamic barrier.

To see whether the kinetics and thermodynamics are closely linked, we compared the kinetic transition times with the corresponding barrier heights. In Figure 4—figure supplement 4B and C, we found strong correlations between the thermodynamic barriers and kinetic rates for both of the transitions between the IS and BS. This observation has demonstrated that thermodynamics and kinetics can be used to infer each other when it is difficult to measure one of the two in practice (Cao, Huang and Liu, 2016; Wang, Martins and Lindorr-Larrson, 2017).

The further quantitative link between thermodynamics and kinetics was investigated by comparing the thermodynamic stability obtained from the Umbrella Sampling simulations and kinetic stability, which is calculated by the logarithmic ratio between the forward and backward rates from the metadynamics simulations (Wang, Martins and Lindorr-Larrson, 2017). From Figure 4—figure supplement 5A, although we observed a strong correlation between these two quantities with different 𝜌, the quantitative link seems only modest (an apparent deviation from the y = x reference line). The significant deviation was observed at the large free energy differences. It may be due to the sampling issue in the thermodynamics simulations when the system shows very distinct bistable populations at the IS and BS separated by a very high free energy barrier, or/and the precision of the kinetics calculations when the transition events are very slow.

We note that there may be another reason that has led to the inaccuracy. From the free energy profiles, we see that the IS has a wider distribution than the BS. This is more obvious when 𝜌 is small. The dwell time at the basin should also take into account the distribution rather than simply using only the lowest free energy points. In principle, the transition time t*_trans_^*^* can be obtained from the double integral of the free energy profiles projected on the reaction coordinate *d_RMS_* (Socci, Onuchic and Wolynes, 1996; Chahine et al., 2007):Ttrans∗= ∫dRMSdRMS(End)d (dRMS) ∫dRMS(Start)dRMSd (dRMS′) exp⁡(F(dRMS)−F(dRMS′)) /kTrd (dRMS)

, where *D(d_RMS_)* is the position-dependent diffusion coefficient. By approximately setting *D(d_RMS_)* to a constant, we calculated the kinetic stability from the free energy profile written as ln(t*_IS->BS_^*^/* t*_BS->IS_^*^*), which was subsequently compared to the “real” kinetic stability obtained from the Metadynamics simulations (Figure 4—figure supplement 5B). In order to be consistent with the Metadynamics simulation, where the transition event was defined as when the binding reached a threshold of *d_RMS_*. We set *d_RMS_(Start)=0.0nm* and *d_RMS_(End)=2.5nm* for the transition from the BS to IS and *d_RMS_(Start)=3.5nm* and *d_RMS_(End)=0.1nm* for the transition from the IS to BS. We see an improvement on the quantitative consistency between these two kinetic stabilities (Figure 4—figure supplement 5B, smaller magnitudes of the slope towards 1 and the intercept towards 0 than those in Figure 4—figure supplement 5A), which were independently calculated from the two different simulations, from that compared with the barrier height (Figure 4—figure supplement 5A). Nevertheless, there is still an element to consider, which we may attribute to the fair treatment for the constant *D(d_RMS_)*. *D(d_RMS_)* has been found to be position-dependent in the binding-coupled-folding process of an intrinsically disordered protein (Chu et al., 2019), though the effect is minor (a cause of ~2kT free energy shift was observed at the binding complex in our previous study with SBMs).

“Direct simulations of the transitions between the IS and the BS are expected to be computationally demanding owing to the high barriers between these two states. […] Further details of the frequency-adaptive metadynamics simulations can be found in SI Appendix 1.”

4) They stated that "We found a monotonic increase of barrier height for both two transitions between the IS and BS as 𝜌 increases (Figure 4E)." Without error estimations, it is hard to judge if the trend for BS→IS barrier increase with 𝜌 is significant or just within the errors. I would strongly suggest they do error estimations and include error bars in the free energy profiles.

We appreciate this suggestion. In order to perform the error analysis, we have run additional three sets of umbrella sampling simulations (there are now a total of four sets of umbrella sampling simulations) with the same simulation parameters except for using different initial conditions (different initial DPO4-DNA configurations and velocities). These combined four independent Umbrella Sampling simulations allowed us to perform the error analysis. As shown in the new figures, all the conclusions made in the original manuscript are preserved. See Figure 4A and Figure 4—figure supplement 4A.

5) There is one experimental author involved in this manuscript, so I read this work as an experimental/simulation collaboration, in which the simulations provide valuable predictions for experimental tests and validations. Besides the comparison with experimental Kd, it will strengthen the work by more comparisons. I understand that it is always non-trivial to combine and compare experiments and simulations, but I will appreciate if they could discuss and suggest the possibilities that could be tested by further experiments.

We thank the reviewer for this comment. We incorporated one paragraph in the Discussion and Conclusions section of the revised manuscript which describes how our theoretical findings may impact the future experimental studies. Please see the Discussion and Conclusions section.

“Our theoretical predictions can be potentially assessed by targeted biophysical experiments. […] The well-developed experimental approaches for DPO4, such as melting circular dichroism (CD) spectroscopy (Sherrer et al., 2012), tryptophan fluorescence (Wong et al., 2008), real-time FRET (Xu et al., 2009; Maxwell et al., 2014; Raper and Suo, 2016), and structural determinations (Ling et al., 2001; Vaisman et al., 2005; Ling et al., 2004b,a; Vyas et al., 2015), can subsequently be used to investigate the kinetics and mechanism of DPO4 folding, DNA binding, and nucleotide binding and incorporation.”

6) Could the conclusions in this manuscript be extended for other multi-domain proteins? Or how general are the conclusions?

This is a good suggestion from the reviewer and we thereby added several sentences in the Discussion and Conclusions section. Please see the Discussion and Conclusions section.

“Our modeling and simulations are applicable to various Y-family DNA polymerases, and we expect similar 1ndings to be obtained for these. [...] Our results can offer useful guidance for protein design and engineering at the multidomain level.”

[Editors’ note: what follows is the authors’ response to the second round of review.]

Revisions:In comparison to the original submission, the reviewers agree that manuscript has been substantially improved in terms of both presentation and substance. The reviewers suggested several relatively minor revisions:Results paragraph four indicates that the ratio of inter and intra-domain contacts was varied. How this was implemented was not clear from the text. For example, one could vary one weight, or the other, or one could modulate both, while keeping some other quantity (e.g. total stabilizing energy) constant. Without this clearly defined, it is difficult fully appreciate the potential significance of any trends based on this ratio.

We thank the reviewers for pointing out this issue. In this study, we modulated the ratio (ρ) of the interdomain strength versus intradomain strength in structure-based models (SBMs) by changing only the strength of interdomain contact (ε_Inter_) while keeping the other parameters in SBMs the same as they are in default. Since the default value of intradomain contact strength in SBMs is 1.0 (ε_Intra_=1.0), here we can simply have ρ=ε_Inter_/ε_Intra_=ε_Inter_. For kinetic simulations, we further rescaled the total potential of the SBM (V_SBM_^apoDPO4^) by multiplying a ratio of *T_f_^ρ0^/T_f_^ρ^* to ensure that the DPO4 folding temperature with different ρ remains the same as the one in default (ρ_0_=1.0) (Please see details in the “Materials and methods” section).

We have added the descriptions of how we practically implemented the modulation of ρ in SBMs in the “Results” section:

“In practice, this is implemented by changing only ε_Inter_ while keeping ε_Intra_ and the other parameters to the default as they are in a homogeneously weighted SBM…”

The authors should expand the discussion on the potential experimental consequence of their conclusions. For example, what signature might single molecule force spectroscopy observe as an implication. What other experiments can be conducted in this regard.

This is a good suggestion. In our previous manuscript, we made a preliminary discussion regarding the potential assessments that can be made in the future experiments to verify our simulation results. Also agreed with this comment, we are now aware that the previous discussion is incomplete and needs further extension with more details. Therefore, we have incorporated several sentences in the discussion of the revised manuscript describing how our theoretical findings may impact future experimental work on this issue.

“Subsequently, the well-developed experimental approaches for DPO4 can be used to investigate the kinetics and mechanism of DPO4 folding, DNA binding, and nucleotide binding and incorporation. […] Furthermore, the structural determination of DPO4 in complex with a damaged DNA substrate and an incoming nucleotide (Ling et al., 2001; Vaisman et al., 2005; Ling et al., 2004b; Bauer et al., 2007; Ling et al., 2004a; Vyas et al., 2015) can provide insights into the effects of backtracking caused by weakening intradomain interactions via mutations, which disrupt the structures of the ternary complexes.”

The manuscript repeatedly describes effects, relative to the "default" parameterization of a structure-based model. It is not clear what significance the default parameters may have. That is, are the default parameters considered to be an accurate approximation to systems in the cell? If so, how, and to what extent? Perhaps the initial parameterization is far from appropriate for the current application, in which case variations in the ratio of different interactions may correspond to a regime that is not biologically relevant. It is important that the authors present these trends in terms of the physical insights into a biological process.

This is a useful comment. In our simulations, we see that the SBM using default parameters (r=r0=1.0) captures many of the experimental observations on DPO4 folding and DNA binding. The consistency is mainly reflected in the following two aspects. (1) The existence of the folding intermediates: Both our current simulations and our previous experiments identified the intermediate states formed during DPO4 (un)folding, which indicates asynchronous (un)folding of individual domains in DPO4 (Sherrer et al., 2012). Structural analysis reveals that there are much more native contacts formed by intradomain than interdomain in DPO4 (Appendix 2—table 1), so the individual domain folding is strongly favored by the default, topology-based SBM, and it is minorly affected by the presence of the other domains. Such a decoupling of the domain folding is encoded in DPO4 native topology, which is the key element for forming the folding intermediates. (2) The complex conformational dynamics of DPO4 during DNA binding: Increasing experimental evidence suggested that DPO4-DNA recognition is a complex three-state binding process accompanied by the conformational transition occurring in DPO4 (Brenlla et al., 2014 ; Raper et al., 2016). Using the default SBM, we obtained an experimentally consistent picture of dynamical DPO4-DNA binding and further characterized a conformational equilibrium shift in DPO4 as DNA binding proceeds. These two features manifest that the mechanisms of the folding and the DNA-binding of DPO4 are largely determined by the native topologies of the protein and the complex, reminiscent of what has been observed in the single-domain protein folding (e. g., Clementi et al., 2000) and the protein-protein binding (e. g., Levy et al., 2004).

However, the in-depth analysis of the results from the default SBM and the measurements from the experiments remains elusive, and sometimes the comparisons lead to conflicting observations. For folding, due to the lack of precision determination on DPO4 (un)folding kinetics, whether DPO4 in reality has further optimized its internal topology-based interactions through assigning the heterogeneity into the intradomain and interdomain interactions to accelerate folding is unknown. Our simulation results suggest that weakening interdomain interactions in the default, homogenously weighted SBM can accelerate DPO4 folding in favor of the “divide-and-conquer” scenario. There are rich hydrophobic residues inside the domains in DPO4 (Ling et al., 2001), which should naturally increase the weight of intradomain interactions in the default SBM. Thus, the deduction from our simulations is seemingly true that the default SBM may have neglected the extra contributions of the intradomain hydrophobic interactions in DPO4 by over-weighting the interdomain interactions to slow down the folding. For binding, there are two prominent inconsistencies between the default SBM and experiments. One is the binding affinity, which has been significantly enhanced by the default SBM. The default SBM quenches conformational flexibility in the DPO4-DNA complex because of the relatively strong interdomain interactions in DPO4, leading to an enhanced binding affinity. The other is the aberrantly stable IS, which corresponds to functionally inactive complex. The default SBM promotes the formation of the IS rather than the functionally active BS, so the subsequent DNA replication process is not favored. Our results show that the weak interdomain interactions in DPO4 not only stabilize the BS, but also facilitate the transition between the active and inactive complex, in favor of DNA binding.

Therefore, we see that the default SBM can in part describe the folding and DNA-binding of DPO4, as both of these two processes are driven by the native topologies of the protein and complex, compatible with the principle of minimal frustration. However, the accurate descriptions require further and detailed parameterizations on the SBM. Our simulations, with a focus on modulating the interdomain interaction strength in DPO4, underlines the importance of weakening interdomain interactions in the default SBM in increasing the folding speed and generating an experimentally consistent picture of the DNA-binding process. We have added several sentences regarding this comment in the “Discussion and conclusions” section:

“We see that the default SBM with homogeneously weighted intra- and interdomain interactions can lead to many consistencies in describing the DPO4 folding and DPO4-DNA binding processes with experiments. […] Considering both the DPO4 folding and the DPO4-DNA binding results, we have suggested that the weak interdomain interactions in DPO4 are the key to the trade-off between DPO4 folding and function.”

The authors may further revise their figures and make them even more intuitive. For example, rho could be labelled as relative strength of inter vs. interactions on the figure, so could MTCI. The labelling of Figure 2G is not very clear, and similar issues exist for some other figure panels.

We have followed this suggestion and modified our figures related throughout the revised manuscript.